# Predictive design of sigma factor-specific promoters

Maarten Van Brempt[1,3], Jim Clauwaert[2,3], Friederike Mey[1], Michiel Stock[2], Jo Maertens[1], Willem Waegeman[2,4] & Marjan De Mey [1,4✉]

To engineer synthetic gene circuits, molecular building blocks are developed which can modulate gene expression without interference, mutually or with the host's cell machinery. As the complexity of gene circuits increases, automated design tools and tailored building blocks to ensure perfect tuning of all components in the network are required. Despite the efforts to develop prediction tools that allow forward engineering of promoter transcription initiation frequency (TIF), such a tool is still lacking. Here, we use promoter libraries of *E. coli* sigma factor 70 ($\sigma^{70}$)- and *B. subtilis* $\sigma^B$-, $\sigma^F$- and $\sigma^W$-dependent promoters to construct prediction models, capable of both predicting promoter TIF and orthogonality of the $\sigma$-specific promoters. This is achieved by training a convolutional neural network with high-throughput DNA sequencing data from fluorescence-activated cell sorted promoter libraries. This model functions as the base of the online promoter design tool (ProD), providing tailored promoters for tailored genetic systems.

---

[1] Centre for Synthetic Biology (CSB), Department of Biotechnology, Ghent University, 9000 Ghent, Belgium. [2] KERMIT, Department of Data Analysis and Mathematical Modelling, Ghent University, 9000 Ghent, Belgium. [3] These authors contributed equally: Maarten Van Brempt, Jim Clauwaert. [4] These authors jointly supervised this work: Willem Waegeman, Marjan De Mey. ✉email: marjan.demey@ugent.be

The flux through a metabolic pathway is not the result of multiple individual enzymatic reactions, with a single rate-limiting step determining the production rate. In concordance with these findings since the late 1990s, the emphasis of metabolic engineering shifted from overexpressing single genes to a more integrated approach that recognizes the importance of gene expression fine-tuning to balance all reactions in a multigene pathway[1–3]. Therefore, enhancing the performance of microbial cell factories, by simultaneously varying genetic parts that determine the expression profile of the heterologous production pathway, has proven a successful strategy[4–7]. More recently, the ability for precise and reliable gene expression fine-tuning has become even more important with the emergence of genetic circuit engineering. Here, whole synthetic biological systems with artificial programs for controlling gene expression are engineered, for which the ability to precisely tune the expression levels of all components in the network is a requisite for functional signal transfer[8–10].

The degree of active enzyme in a cell can be tuned on multiple levels of control, i.e., transcriptional, translational, and enzymatic levels. Numerous genetic building blocks have been developed to adjust these levels by taking advantage of different properties of the cell's machinery[9]. In prokaryotes, the synthesis rate of RNA and subsequent protein can be directly regulated by altering the DNA sequence of the promoter and ribosome binding site (RBS), which determine their affinity for, respectively, the RNA polymerase and ribosome[11,12]. Also, synthesis rates can be modulated by using various (transacting) transcriptional and translational activators or repressors, such as LacI variants, STARs, dCas9/gRNAs, and sRNAs[13–16]. Other strategies include controlling the degradation rates of RNA and protein, e.g., by targeting their stability, or enhancing the pathway's catalytic efficiency by building synthetic scaffolds[17–19].

The rapidly expanding toolbox of characterized expression control elements contributes to the continuous extension of engineering possibilities and allows the computational design of genetic networks[20–22]. However, a defined set of parts inherently limits the flexibility with which genetic designs can be created. The ability for exact fine-tuning, imposing user-defined part sequence constraints or building functional degenerated part libraries requires a deeper sequence—function relationship understanding and tools to predict the parts' behavior. Various in silico models that facilitate the de novo design of different expression control elements have been developed. These include methods to achieve tailored gene expression directly, such as promoter transcription initiation frequency (TIF) engineering[11,23–31] and tuning of the RBS properties[12,32–36]. Indirect approaches include the engineering of mRNA stability through transcriptional terminators[37] and the synthetic regulation using dCas9[38] or various riboregulators[39–42]. Furthermore, the built models are the basis for the development of several (online) tools allowing user-specific de novo sequence design[12,35,41–44]. The synthetic biology community's demand for readily applicable forward engineering tools is, for example, expressed by the success of the RBS sequence design tools created by the Salis Lab. Up to October 2020, their algorithms have been used to design over 620,000 genetic sequences by over 9100 users worldwide, including 24 industrial licensees[45].

To date, despite the abundance of work on promoter engineering and modeling, a standardized and readily applicable tool is not available for de novo promoter design. However, transcription being the first step in gene expression, promoter tuning is most essential. It allows e.g., the tuning of noncoding RNA abundance, which can fulfill various regulatory functions, or plays an important role in combination with RBS strength in the economy of gene expression[46,47].

A prokaryotic promoter can generally be subdivided in several defined parts that determine the interaction with the RNA polymerase (RNAP), and thus its promoter functioning, offering multiple strategies for promoter engineering[48–50]. Change in promoter TIF has been generated by modifying (i) the conserved −35 and −10 regions, which actuate the selective recognition and interaction by the sigma factor ($\sigma$) RNAP subunit[30,51–53], (ii) the UP element, situated upstream of the −35 conserved region and interacting with the αCTD RNAP subunits[29,54], (iii) the spacer sequences, spanning the region between −35 and −10 elements and upstream of the −10 element[55–58], or (iv) a combination of multiple, or all, of these elements in a random[11,27,59,60], or modular manner[24,53,61].

Previous attempts to model promoter strength in function of its DNA sequence have often targeted multiple of these promoter regions simultaneously, severely underestimating the complexity of interplay between regions, and/or employed modeling methods that assume independence between mutations (e.g., position weight matrix), practically limiting their predictability to single nucleotide variations[11,23,25–28,30,31]. These factors, and a substantial lack of data to grasp the promoter's structural complexity or support more complex models, often resulted in weak correlations or low promoter strength discrimination resolution. Conversely, some work indicates that the promoter's −35 and −10 conserved regions are the greatest determinants of promoter TIF and could lead to predictions with high accuracy[24]. However, these relatively short sequences offer small sequence flexibility, resulting in large sequence repeats when using multiple promoters. More importantly, changes in these regions abolish the promoter's ability to selectively bind specific $\sigma$ RNAP subunits. As the orthogonality of gene expression in synthetic genetic systems becomes increasingly important, the −35 and −10 conserved regions are to be left unchanged to ensure this property in the design of genetic circuits[50,56].

In previous work[56], to preserve orthogonality of the promoter sequence with specific $\sigma$s, promoter libraries were constructed by randomizing the promoter spacer nucleotides spanning between the −35 and −10 conserved regions, resulting in a five log range of promoter TIFs. Here, we build upon this work by combining the use of fluorescence-activated cell sorting (FACS) on these libraries and targeted high-throughput DNA sequencing to obtain considerably large data sets (250,000–400,000 unique sequences per setting) holding promoter sequence–function relationship information in view of predicting promoter TIFs, and designing promoter sequences with a specific TIF. A computational model was trained on these newly created data, to develop the first in vivo validated Escherichia coli (E. coli) $\sigma^{70}$ Promoter TIF Designer tool, named ProD. Our tool is able to output the promoter spacer sequence, constituting 17 variable consecutive nucleotides. Additionally, albeit lacking in vivo validation, predictive models for promoter strength and orthogonality have been trained and evaluated to expand such a tool with three different promoter architectures with specificities toward heterologous Bacillus subtilis (B. subtilis) sigma factors B, F, and W. This is achieved by the unprecedented size of the characterized promoter libraries and the use of convolutional neural networks, a machine learning methodology achieving state-of-the-art performances on motif detection tasks[62]. The neural network was adapted for ordinal regression, a classification problem for values with a meaningful ordering (i.e., ordinal values), by applying constraints to the output nodes of the model.

## Results

**Promoter TIF pooling and genotyping.** In previous work, we have demonstrated the use of chimera RNA polymerases based

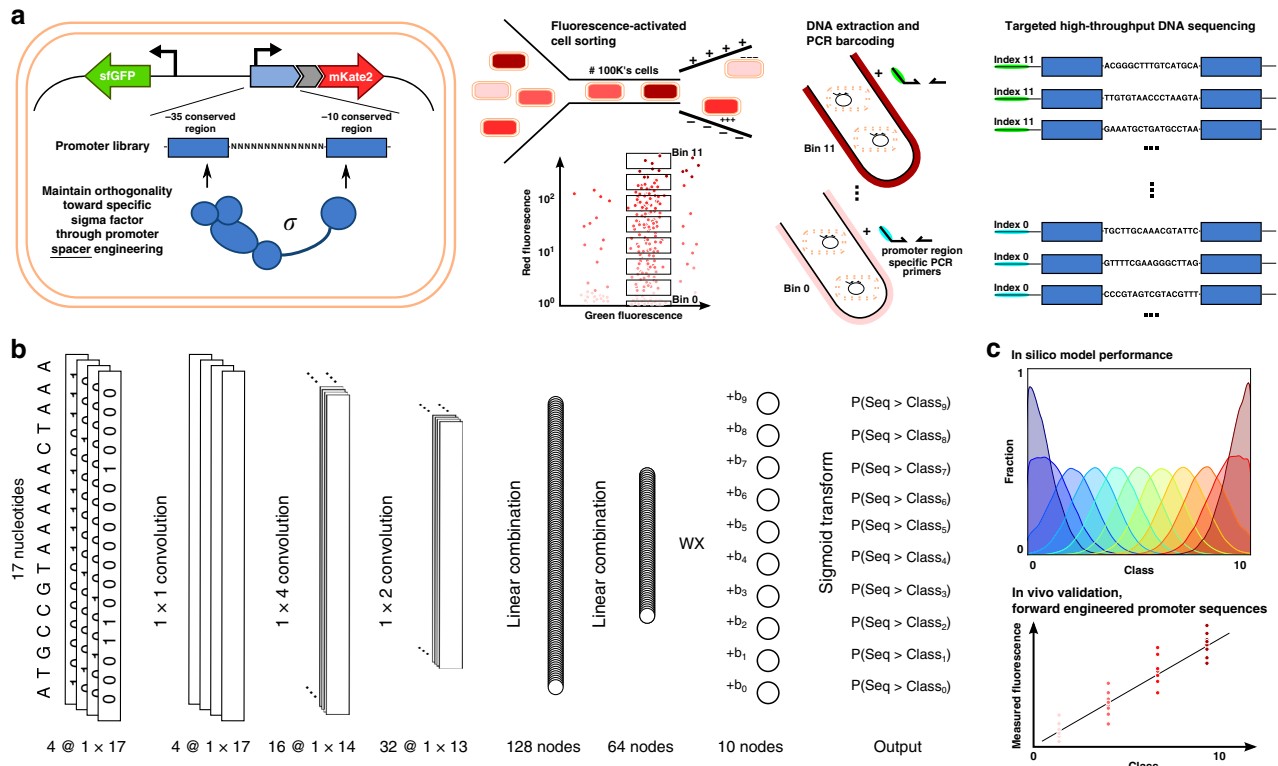

**Fig. 1 Illustrative overview of the workflow to create our Promoter transcription initiation frequency (TIF) Designer tool, ProD. a** Multiple promoter libraries were created, driving mKate2 expression, by engineering the promoter spacer DNA sequence to maintain the sigma factor recognition specificity. Cells were sorted by fluorescence-activated cell sorting in 12 separate bins, according to the level of red fluorescent protein expression. Subsequently, plasmid DNA was extracted and the promoter regions were amplified and barcoded uniquely for each bin. High-throughput DNA sequencing was used for genotyping. **b** The architecture of the neural network for ordinal regression trained on the created data sets for promoter TIF. Seventeen nucleotide sequences are processed by four 1 × 1, 16 1 × 4, and 32 1 × 2 convolutions and two fully connected layers of 128 and 64 nodes. A latent variable, correlated to the TIF of the promoter, is obtained through a single linear combination of weights (**w**) with the 64 output nodes (**x**). A vector of ordered biases **b**, optimized during training, outputs ten shifted values relative to the latent variable. The sigmoid transform of these outputs represents the probability of the TIF of the sequence being greater than a given class y. **c** The model with a minimum loss on the validation set is selected and evaluated on the test set, showing the ordinal correlation between its predictions and the true classes. By random sampling, a set of promoter sequences is generated and a selection is made, as predicted to display a range of promoter TIF levels covering the different classes (0–10), for in vivo validation (only performed for the *E. coli* σ⁷⁰ promoter TIF model).

on heterologous σs from *B. subtilis*, that recognize specific promoter sequences to create a functional and orthogonal expression system in *E. coli*. To this end, the spacer sequence between the conserved −35 and −10 σ recognition sites of *E. coli* σ⁷⁰ and *B. subtilis* $\sigma^B$, $\sigma^F$, and $\sigma^W$-specific promoter sequences (17 bp, 12 bp, 15 bp, and 16 bp, respectively) was engineered, to introduce variability in promoter TIF while preserving the orthogonal features toward specific σs[56]. A vector (pLibrary) was constructed for each σ-specific promoter, consisting of a promoter library site in a red fluorescent protein (mKate2[63]) expressing operon, and a second operon, constitutively expressing sfGFP[64] as an internal reference for normalization (Fig. 1a). In this work, to define the spacer sequence–function relationship, first, FACS was used to sort cells, in accordance to cellular fluorescence (proxying promoter TIF), into 12 sorting bins on all four promoter libraries. Next, high-throughput DNA sequencing of each bin was performed for promoter genotyping (Fig. 1a). Additionally, the vectors containing the different promoter libraries were cloned into strains containing their noncognate σs in the genome. These cells were sorted into a nonfluorescent and fluorescent subpopulation, indicating conservation or loss of orthogonality, respectively. Similarly, through the use of high-throughput DNA sequencing, genotypic data was acquired. For cell sorting, constitutive sfGFP expression was taken into account to exclude

artefacts or cells showing aberrant expression. Also, to reduce the overlap between different levels of promoter TIF, considering the inherent Gaussian character of the expression profile of a single promoter sequence, buffer regions have been included between adjacent bins. The different bins account for a maximum of 20% to under 0.01% of the population size, ensuring the acquisition of genetic information of underrepresented promoter TIFs, such as seen for the lowest and highest rates of all libraries (see Fig. 2). All cell sorting schemes are depicted in Supplementary Figs. 1–4. Sorted bins were cultured and the plasmid DNA was isolated. The promoter region was amplified and tagged with bin-specific indexes for subsequent high-throughput sequencing, resulting in a total of ca. 9,000,000 reads, holding information to link promoter sequences to their expression level.

**Data selection methods to map ambiguous read sequences**. For our *E. coli* σ⁷⁰ specific promoter library, a total of 1,386,614 reads were obtained, covering 321,575 (23.19%) unique sequences. From the unique sequences, 211,772 (65.85%) have more than one read, with 117,249 (36.46%) instances having reads sequenced in multiple bins. To identify possible inconsistencies within the data, several data properties (see "Materials and methods" section) have been derived from sequences featuring multiple reads: the total amount of reads, the number of bins with

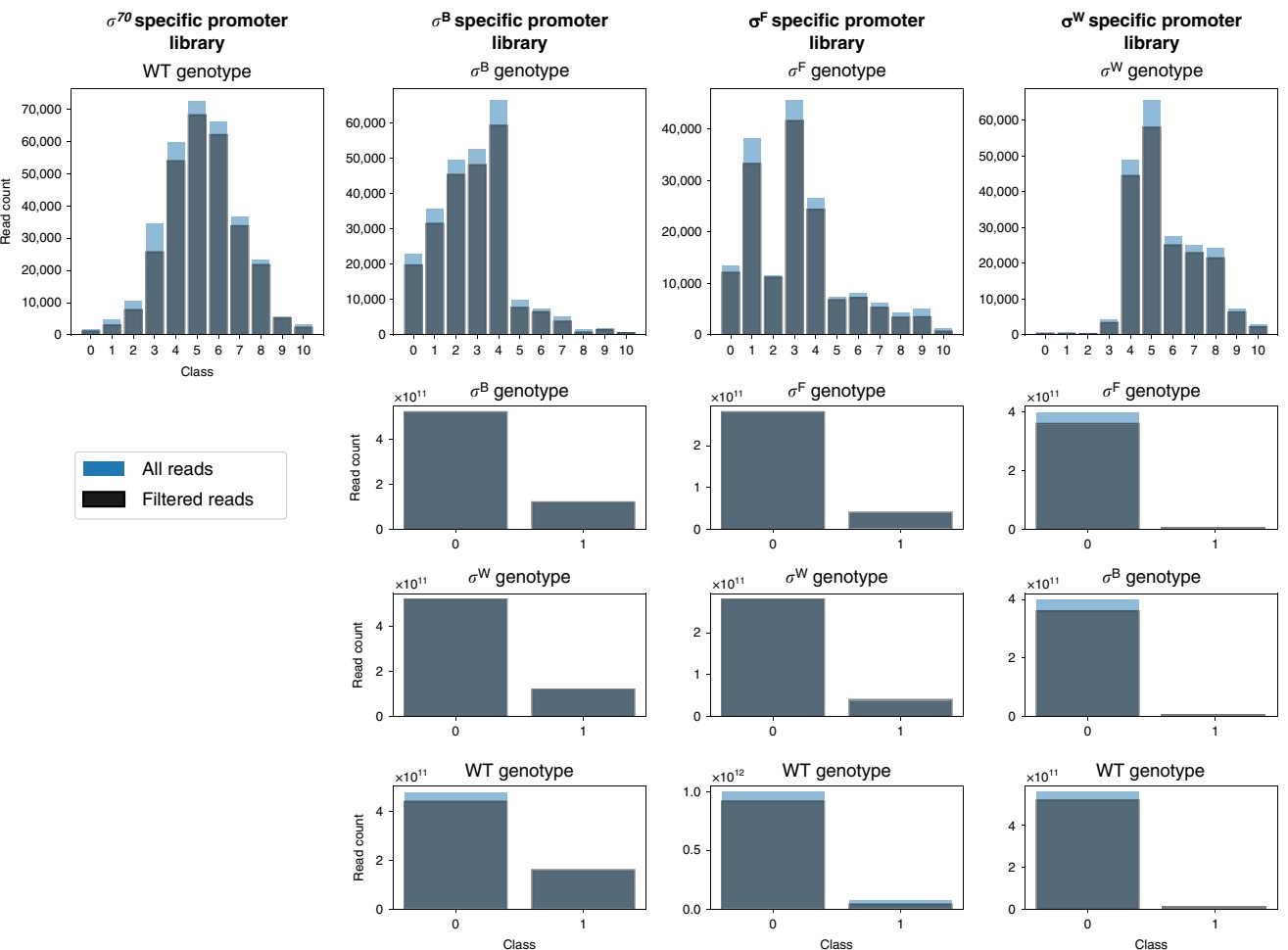

**Fig. 2 An overview of the data available for each setting in relation to the labeled sequences used for the prediction task.** For each setting is given: the total number of sequences and the part used for model training after exclusion of the outliers. For each promoter library the genotypic background is given: heterologous sigma factors ($\sigma$) B, F, or W from *Bacillus subtilis* expressed in *Escherichia coli* (*E. coli*), or wild-type (WT) *E. coli* harboring no heterologous sigma factors. For the binary setting, the positive class represents fluorescence, indicating the loss of orthogonality. An overview of the data samples for each setting is listed in Supplementary Table 1. Source data are provided as a Source Data file.

the highest amount of reads and their relative distances. A plethora of sequences have reads present in bins that provide incompatible properties, such as the presence of reads in distant bins constituting both low and high expression rates. Therefore, in order to exclude aberrations in the data, promoter sequences with outliers (5% highest values) on any of the data properties were removed. Furthermore, reads recovered from bin 0 (for promoter libraries sorted in presence of their cognate sigma factor) have not been considered for training. This bin covers all sequences with background levels of red fluorescence. Absence of functional mKate2 expression might be caused by a variety of factors, such as mutations outside the promoter region or other cell defects. A preliminary analysis showed the exclusion of this bin to result in better accuracy of the model on the test set. This is not surprising, as the relative small sequence count and marginal position of the bin results in a high impact on the loss function of the ordinal regression setting, as a result of weighing the loss in function of the bin size (see "Materials and methods" section). The *E. coli* $\sigma^{70}$ promoter TIF prediction model was trained on 284,421 (88.45% of total) unique sequences covering 938,863 (67.70% of total) reads. For training purposes, the label assigned to each sequence is the bin with the most reads. As bin 0 has not been included to train the model, sequences from bin 1 through 11 are adopted as classes 0 through 10. An overview of the

amount of labeled sequences for each class before and after data filtering is depicted in Fig. 2. An extended overview of the recovered data is given in Supplementary Table 1.

**Exploratory sequence analysis.** In previous studies attempting to model promoter TIF in function of its DNA sequence, one or more nucleotides at specific positions could often be identified as significant determinants of the TIF[11,23,25–27,30,31]. Prior to model building, we created sequence motifs of the promoter spacers, for every class (~promoter TIF), to detect potential single nucleotides that dominantly determine promoter TIF. The motifs are depicted in Fig. 3.

In general, the majority of the motifs do not show any specific nucleotides with higher occurrence frequencies, indicating an equal distribution of the four nucleotides at any given sequence position. The motifs that do show up, comprising half the spacer ($\sigma^B$, $\sigma^F$, $\sigma^W$) or the three-nucleotide sequence "TCG" ($\sigma^{70}$), adjacent to either the −35 or −10 conserved regions, can be explained by the architecture of the libraries[56]. Given motifs are overrepresented as an effect of the blueprint used for the randomized sequences, thereby showing a substantial presence in the data sets. These specific motifs, with an increased presence in bins representing either higher or lower promoter TIF, resemble

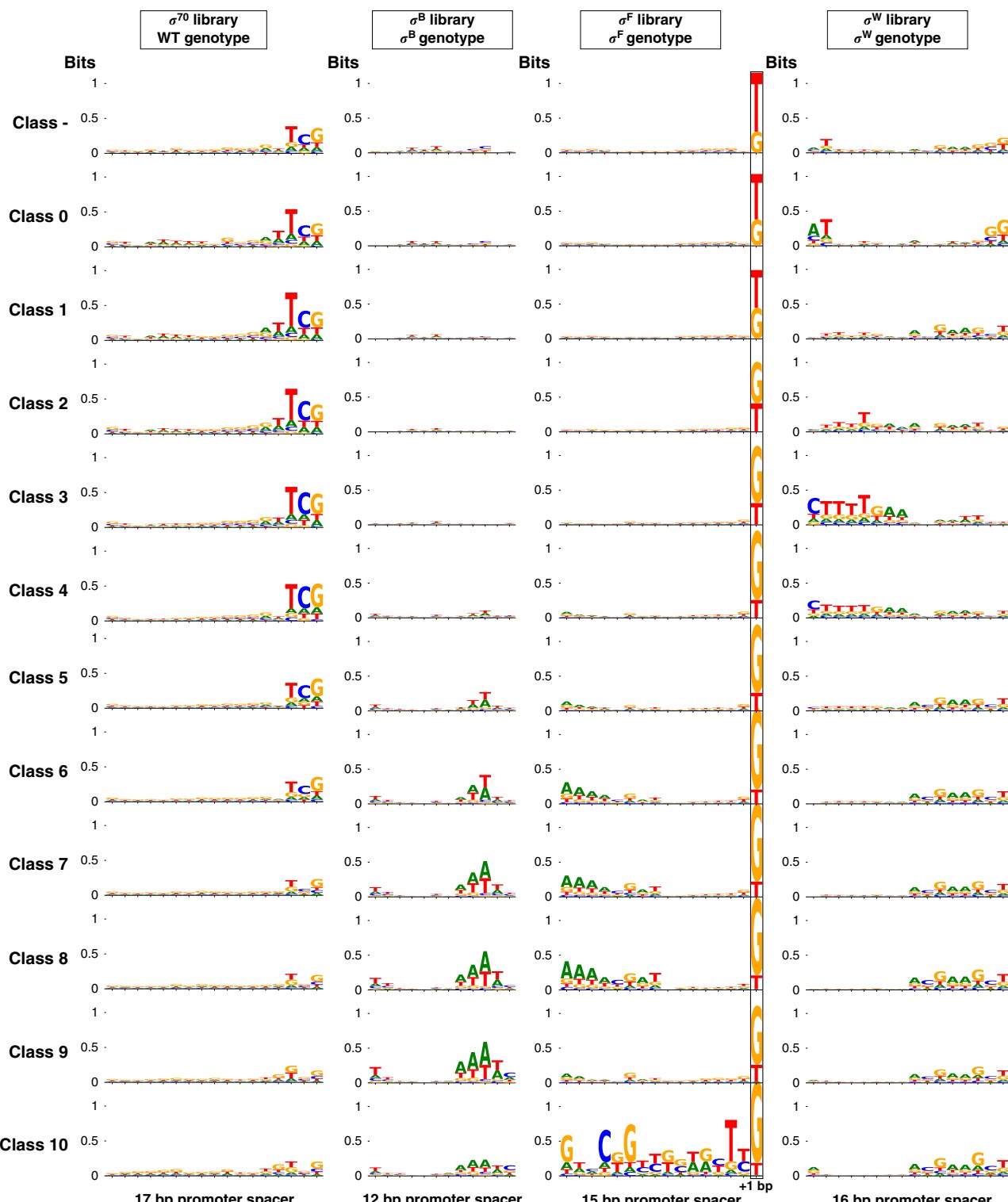

**Fig. 3 Sequence logos illustrating the promoter spacer motifs, situated between the conserved −35 and −10 regions.** These are given for each of the bins in which the *Escherichia coli* sigma factor 70 (σ70) and *Bacillus subtilis* σB, σF, and σW-specific promoter libraries, in presence of their cognate σ, were sorted. Classes 0–10 represent the promoters ranging from the lowest to the highest expression. In case of the σF library, one engineered base pair following the spacer is situated within the −10 conserved promoter region[56]). Motifs were created with WebLogo3.6.0[83], using the filtered data sets (WT wild-type).

**Table 1 Performance metrics of the models on the test sets.**

| | $\sigma^{70}$ specific promoters | | $\sigma^B$ specific promoters | | $\sigma^F$ specific promoters | | $\sigma^W$ specific promoters | |
| --- | --- | --- | --- | --- | --- | --- | --- | --- |
| | WT genotype | | $\sigma^B$ genotype | | $\sigma^F$ genotype | | $\sigma^W$ genotype | |
| | Mean | Std. | Mean | Std. | Mean | Std. | Mean | Std. |
| Spearman's rho | 0.574 | 0.003 | 0.565 | 0.002 | 0.497 | 0.002 | 0.234 | 0.050 |
| Weighted ACC | 0.230 | 0.005 | 0.230 | 0.005 | 0.210 | 0.004 | 0.136 | 0.015 |
| Weighted MAE | 1.609 | 0.007 | 1.652 | 0.017 | 1.919 | 0.052 | 2.504 | 0.100 |
| *MAE* | | | | | | | | |
| $y = 0$ | 2.866 | 0.150 | 2.397 | 0.068 | 2.891 | 0.107 | 4.797 | 0.240 |
| $y = 1$ | 1.845 | 0.105 | 1.630 | 0.070 | 2.097 | 0.117 | 4.022 | 0.141 |
| $y = 2$ | 1.372 | 0.054 | 1.071 | 0.028 | 1.507 | 0.101 | 2.739 | 0.287 |
| $y = 3$ | 1.284 | 0.058 | 1.197 | 0.035 | 1.294 | 0.056 | 1.613 | 0.329 |
| $y = 4$ | 1.241 | 0.045 | 1.368 | 0.058 | 1.324 | 0.095 | 1.090 | 0.122 |
| $y = 5$ | 1.324 | 0.023 | 1.407 | 0.055 | 1.357 | 0.049 | 0.587 | 0.480 |
| $y = 6$ | 1.401 | 0.038 | 1.333 | 0.036 | 1.258 | 0.068 | 0.951 | 0.111 |
| $y = 7$ | 1.362 | 0.045 | 1.315 | 0.051 | 1.572 | 0.119 | 1.790 | 0.189 |
| $y = 8$ | 1.316 | 0.055 | 1.507 | 0.065 | 2.102 | 0.146 | 2.711 | 0.246 |
| $y = 9$ | 1.533 | 0.032 | 1.730 | 0.116 | 4.010 | 0.135 | 3.774 | 0.205 |
| $y = 10$ | 2.158 | 0.047 | 3.224 | 0.165 | 1.704 | 0.320 | 4.646 | 0.298 |
| | | | $\sigma^F$ genotype | | $\sigma^B$ genotype | | $\sigma^B$ genotype | |
| ROC AUC | | | 0.694 | 0.004 | 0.652 | 0.004 | 0.615 | 0.010 |
| | | | $\sigma^W$ genotype | | $\sigma^W$ genotype | | $\sigma^F$ genotype | |
| ROC AUC | | | 0.691 | 0.002 | 0.643 | 0.004 | 0.635 | 0.006 |
| | | | WT genotype | | WT genotype | | WT genotype | |
| ROC AUC | | | 0.665 | 0.004 | 0.632 | 0.003 | 0.635 | 0.006 |

Weighted metrics are used for accuracy and mean absolute error to account for class imbalance. For each performance, the mean and standard deviation (Std.) are given obtained by training multiple models in a five-fold set-up for the test set. Mean absolute errors for each of the sample classes ($y = 0$–10) are given. ROC/PR AUC is used for the binary classification problem. ROC AUC represent a perfect model at AUC = 1. (*ACC* accuracy, *MAE* mean absolute error, *AUC* area under the curve, *ROC* receiver operating characteristic, *WT* wild-type, $\sigma$ sigma factor). Source data are provided as a Source Data file.

the observations made in our previous work, where the different libraries were characterized using flow cytometry[56]. With respect to the $\sigma^B$ library, the higher presence of the "A" and "T" nucleotides near the −10 conserved region, with exception of the base pair most adjacent to this region, indicate the importance of a local low GC% in enhancing promoter TIF. In case of $\sigma^F$ libraries, libraries were constructed allowing a "G" nucleotide besides the original "T" in the −10 conserved region most adjacent to the spacer. An increased presence of "G" in the bins representing stronger promoters is observed, confirming literature stating the $G_{-16}$ strongly facilitates $\sigma^F$ recognition[56,65–67]. The strong sequence motif for bin 11 of the $\sigma^F$ library is most likely due to the relatively small number of unique sequences in the data set (486 sequences).

Sequence motifs were also created for the libraries sorted in presence of their noncognate sigma factors, indicating the promoters' state of orthogonality (Supplementary Fig. 5). No indications of single nucleotide positions significantly contributing to orthogonality could be detected.

In contrast to the −35/−10 conserved regions and the UP-/extended −10-element, no studies have been published on the interactions of specific base pairs within the spacer with the cell's transcription machinery to this day. The contribution of spacer DNA to promoter TIF is most likely owed to structural features. For example, the length of the spacer is believed to be of importance to correctly align the −35 and −10 conserved promoter regions with RNAP for an efficient recognition[68]. Also, both Liu et al.[57] and Urtecho et al.[24] determined a negative correlation between spacer GC content and promoter TIF. The GC content of our libraries was analyzed for the ten promoter TIF classes in this study (Supplementary Fig. 6). With no linear correlations found, these previous findings are not supported for the four studied promoter chassis.

**Ordinal regression successfully maps the promoter TIF to a latent variable**. To maintain practical feasibility while exploring an enormous sequence space, organisms containing engineered promoters with differing sequences but generating a similar fluorescent signal were grouped together using FACS. Given the amount of data and the ordinal nature of the prediction task, a shallow convolutional neural network adapted for ordinal regression has been created and trained for each sigma factor/promoter data set (Fig. 1b, c). See "Materials and methods" section for a full description of the model architecture and training. The model performances are given in Table 1. Weighted performance metrics are used to make up for the imbalance of sequence counts between the classes, thereby ensuring the importance of each class independent of the total amount of samples it contains. This equals to the unweighted results in the case of a balanced sample distribution. For each class separately, the mean absolute error is given, revealing the uncertainty of the model on class level. This is visually expanded upon in Fig. 4, where the fraction of predicted labels for the samples of each class are given.

Results are shown on the training and test set. The ordinal nature of the model, adapting the use of a single latent variable to distinguish between classes, results for the $\sigma^{70}$ and $\sigma^B$ libraries in a clear ordinal correlation between the true and false positives, the latter of which are distributed around the former. The overlap of predicted classes for neighboring labeled samples is an expected effect of the noise on the labels, introduced by the overlapping ranges of the promoter expression profiles. Models predicting the

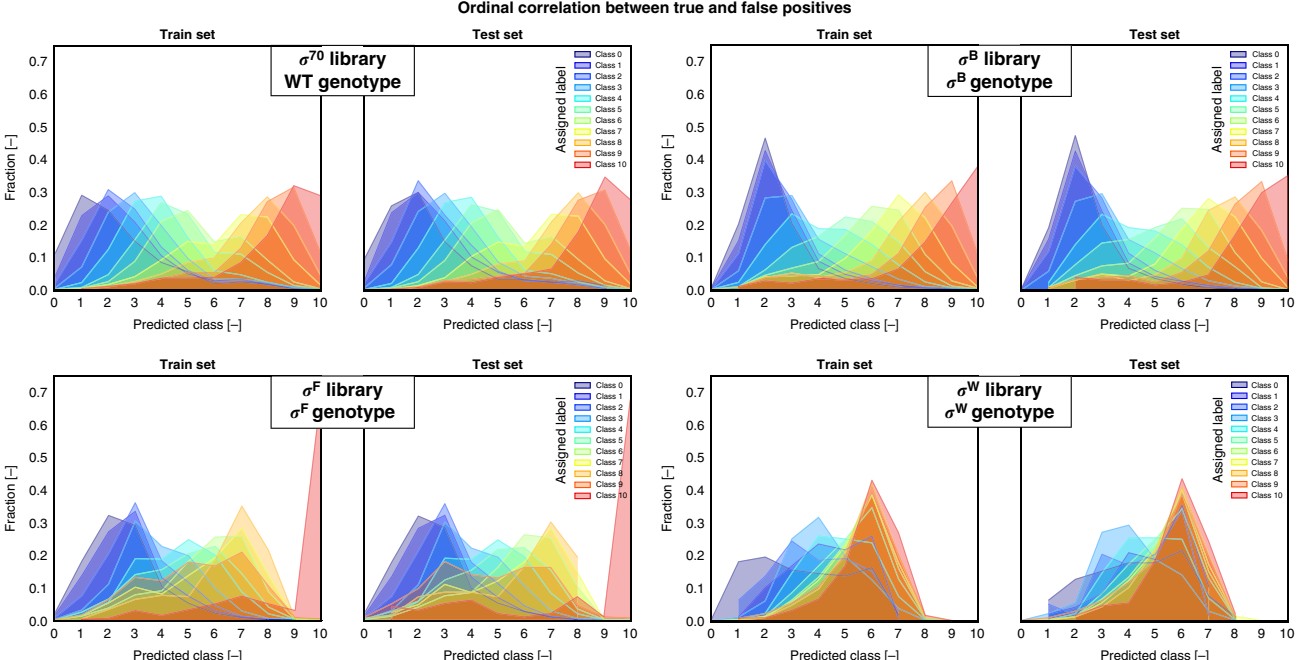

**Fig. 4 Distributions of the predicted class labels for each promoter library on the train (left) and test (right) set.** Different distributions are separated by the true class labels of the sequences. For the test sets, the weighted mean absolute error, weighted accuracy and Spearman correlation are given in Table 1. The distributions, which are centered around the true class labels, demonstrate the advantage of using a model for ordinal regression. The overlap of the different distributions can at least partially be attributed to the expected intrinsic and extrinsic noise on the labels. (σ sigma factor, WT wild type). Source data are provided as a Source Data file.

promoter TIF of σ$^F$-specific and especially σ$^W$-specific promoters have relatively low performances (Table 1), resulting in a less clear ordinal character of the data, with a higher overlap between the distributions of the predicted labels from different classes. Using DeepLIFT[69], a sensitivity analysis of the input as a function of the output was performed on the trained model of each σ-specific promoter library (Supplementary Figs. 8–11). This is done on all sequences in the test set. Attribution scores are calculated based on the gradient (backpropagated from the target class) and signify the relevance of the input nucleotides on the output class prediction.

**In vivo validation of the promoter designer tool (ProD) for the prediction of promoter TIF.** The model that is trained to predict TIFs of σ$^{70}$-specific promoters in ordinal classes was subsequently subjected to in vivo validation. The σ$^{70}$ model was selected due to its relevance for the scientific community. A list of random promoter spacer sequences was generated via randomized sequences and ordered using the trained model. A subselection of 54 promoter sequences was made, predicted to cover the whole library expression range (class 0–10). An additional class was created (denoted as "high") that includes the sequences predicted as class 10 with the highest probabilities in the randomly generated set. Generated spacer sequences were selected to not be present in the original data set, and range from 2 to 5 single nucleotide mutations from the nearest sequence (hamming distance[70]). Supplementary Table 2 lists all the selected spacer sequences with their predicted class and hamming distance towards the closest match in the original data. These were cloned in the pLibrary vector by insertion in the promoter chassis used for library creation. The 54 strains, each containing one of the promoters, were grown in eight biological replicates. After reaching the stationary growth phase, the OD and the fluorescence generated by mKate2 and the constitutively expressed sfGFP were measured and the corrected fluorescence (see

"Materials and methods" section) was calculated. These calculated values are depicted in Fig. 5 as log-transformed values, normalized between 0 and 1. An overview of all measured data is represented in a barplot in Supplementary Fig. 7. The corrected fluorescence values are additionally listed in Supplementary Table 2. The Spearman's rank correlation factor is 0.909 (p-value < 2.2e−16), indicating the ordinal character of the observations with respect to the predicted class. Figure 5 also shows the linear regression fit ($R^2 = 0.876$), together with the 95% confidence interval (shaded area) for the class means and a 50% prediction interval (red dashed line).

Additionally, as a preliminary indication of the in vivo performance of model predictions for σ$^B$, σ$^F$, and σ$^W$-specific promoters, we predicted the TIF class and orthogonality of a limited set of library promoters were constructed and characterized in our previous study[56]. The tested promoters do show an ordinal relation between measured expression level and predicted class, especially for the sigmaB-specific promoters, though the number of observations is small and a fraction of the sequences was also present in the model training sets (Fig. 6). Also, the promoters with loss of orthogonality in vivo showed high probability values for predicted loss of orthogonality. An overview of the promoter sequences and predicted data is presented in Supplementary Table 3.

## Discussion
Despite the development of protein–DNA interaction models enabling the prediction of transcription factors' affinity for DNA (e.g., FeatureREDUCE[71]), or the abundance of studies on correlating sequence features to the rate of transcription, a thorough understanding of the promoter sequence is absent. Furthermore, a promoter TIF prediction tool supporting the de novo design of complete promoters remains absent from the synthetic biologist's toolbox. The progress made on computational methods for motif detection tasks joined with the construction of promoter libraries

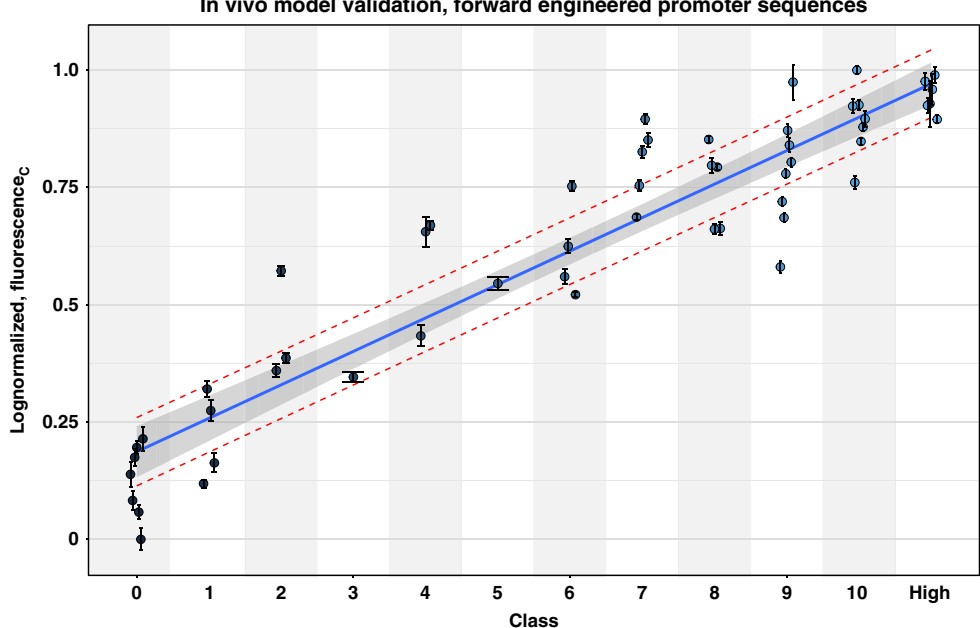

**Fig. 5 Promoter transcription initiation frequency (TIF) of 54 forward engineered sigma factor 70 (σ70) promoters in function of their predicted class.** Values are corrected fluorescence measurements (fluorescence$_C$), logarithmic transformed and normalized between 0 and 1. Data are represented as mean values $+/-$ the standard deviation derived from eight biological replicates. The linear regression line ($R^2 = 0.876$) is depicted together with the 95% confidence interval (shaded area) and the 50% prediction interval (red dashed lines). Source data are provided as a Source Data file.

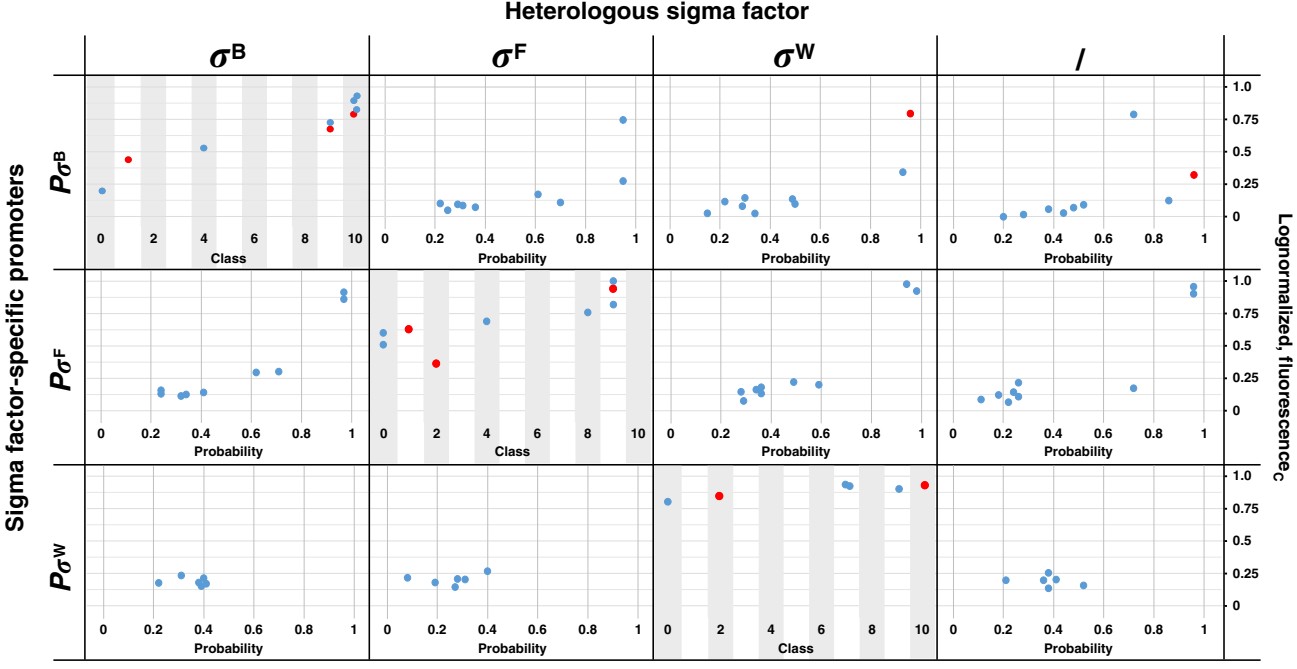

**Fig. 6 Promoter transcription initiation frequency (TIF) of the defined sets of promoters created and characterized in previous work[56] in relation to their predicted class and orthogonality.** Values are corrected fluorescence measurements (fluorescence$_C$), logarithmic transformed and normalized between 0 and 1. Promoters containing deviant spacer lengths were excluded. Higher probability values indicate a higher probability for loss of orthogonality. Data points in red indicate the specific sequence was part of the training data set. Source data are provided as a Source Data file.

of unprecedented size empowered us to take a large step forward in meeting this need. In this work, four models were created to enable forward engineering of *E. coli* σ70 and *B. subtilis* σB, σF, and σW-specific promoters based on their TIF and and mutual orthogonal functioning in *E. coli*. Moreover, the *E. coli* σ70 model was validated in vivo to create the first Promoter TIF Designer tool, named ProD, allowing 17 degrees of freedom, encompassing

the complete spacer that spans the region between the −35 and −10 conserved promoter elements.

The promoter libraries were designed by randomizing only the spacer DNA sequence delimited by the conserved −35 and −10 regions to preserve specific σ recognition, an essential feature in orthogonal genetic circuits[56]. Considering the vastness of the entire sequence space (e.g., $4^{17}$ sequences for the σ70 promoter

spacer), which is practically unfeasible to fully characterize, library promoter sequences displaying similar promoter TIFs were pooled using FACS for subsequent genotyping. The application of a custom built convolutional neural network for ordinal regression proved succesful in capturing the complex interactions, within the promoter spacer sequence in function of promoter TIF. The $\sigma^{70}$ model achieved a Spearman correlation of 0.56 with a mean error of 1.6 classes on predictions. However, in vivo validation on 54 forward engineered promoters resulted in a Spearman correlation of 0.91, indicating a strong ordinal relationship between predicted an measured promoter TIFs.

A notable difference exists between the performance on the test and in vivo validation set. This difference can be explained by the noise present in the FACS data used to train and validate the model. Importantly, binning is performed solely in correlation to the fluorescence of the expressed mKate2 protein, given a minimal expression of sGFP (Supplementary Figs. 1–4), and does not account for extrinsic noise[46,72]. This includes the variability in the expression profile introduced by the environment, where the normalization of the constitutively expressed fluorescence protein accounts for factors such as plasmid copy number and cell size. The presence of both extrinsic and intrinsic noise negatively impact accuracy on the test set. In contrast, measured expression rates of the in vivo validation promoters have been adjusted for extrinsic noise, and performances obtained are therefore expected to better represent the capability of the model.

A higher complexity between the classes is possible for models where no ordinal regression between the classes is enforced. An ordinal relationship of the predicted output is expected, as outlined by the applied FACS procedure. However, due to the high capacity of neural networks and the aforementioned noise present within the data, predictions of given models can result in non-sensible and impractical predictions, such as similar probabilities for distant classes and dissimilar probabilities for bordering ones. Introduction of a single latent variable to project all classes offers an elegant solution at the cost of a lower complexity of the output class. Nonetheless, the substantial increase of the spearman rho between the test and in vivo validation set support the presumption that both the ordinal layer and early stopping (using the validation set) regulates the model against overfitting on noise.

Compared to the $\sigma^{70}$ model, the $\sigma^{W}$ model substantially underperforms and the distributions of the predicted classes for the sequences of each assigned class (Fig. 4) display a clearly less ordinal character, with multiple assigned classes showing the majority of their predictions in the same prediction class. This can be aligned with the observation made in previous work[56], indicating that the capability of introducing TIF variability through spacer sequence engineering is limited, probably due to the structural differences of extracytoplasmic function (ECF) $\sigma$s to which class of sigma factors $\sigma^{W}$ belongs. The model might correctly predict the true mean expression of promoters found in different bins, to be similar. An in vivo validation experiment is essential to support this hypothesis.

In case of the models predicting orthogonality between non-cognate promoter–sigma factor pairs, the performance is presumably limited by the quality of the data present in bin 0, which covers all sequences with either minimal or lack of expression. The abscence of functional mKate2 expression might be caused by a variety of factors that are not related to the promoter sequence, such as cell defects or genomic alteration through random mutations. Data from the bin covering this expression space was excluded to build the promoter TIF models for aforementioned reasons, resulting in a substantial increase in performance. However, this bin is essential in solving the binary classification problem. We made observations indicating that

models for prediction of orthogonality could be enhanced by improving the selection of the lower bins, ensuring the exclusion of aberrant cells, but more in depth research is required to assess the potential reduction of robustness of the model performances. Nevertheless, the predictions on the orthogonality provided by the models on the previously-created set of library promoters[56] are very promising, as the sequences with the highest probability for loss of orthogonality have been reviewed as such (B9 and F8-9 in Supplementary Table 3).

Determination of the features playing the most important role in promoter TIF remains unsolved. Sequence motifs for the different classes of TIF were created to detect contributing nucleotide positions for each class. The absence of pronounced occurrence frequencies of specific nucleotides in the sequence motifs clearly shows that more complex interactions between base pairs are responsible for the observed TIF. Essentially, this is demonstrated by the effectiveness of neural networks, capable of complex higher-order interactions, to evaluate and classify sequences. In contrast, PWM represent the relative frequencies of the nucleotides at every position of the genome, and are therefore incapable of capturing higher order interactions. This also indicates that caution should be exercised with models predicting the effect of single base mutations on promoter TIF (e.g., Meng et al.[31]). Insights from reported observations presumably apply solely to the promoter (spacer) under study and cannot be extended to other base promoters. Rather than by specific nucleotides, it is believed that the spacer sequence influence on promoter TIF is determined on a structural level[48]. We therefore analyzed the correlation between the GC-content of the spacer and promoter TIF, previously established as an inverse correlation, and found our results do not support these observations (see "Exploratory sequence analysis" section). The geometry of base pair steps and shaping of the DNA 3D-structure of a specific sequence is further characterized by a large set of properties (e.g., shift, slide, tilt, and roll). The development of methodologies to study these structural properties for big data could be the key to provide fundamental understanding of the role of the promoter spacer.

Moving beyond the promoter's spacer, the UP element, spanning the region from −40 to −60 and the further upstream promoter background sequence are important factors to take into account regarding promoter TIF. Rhodius et al.[29] showed that the contribution of the core promoter and UP elements to the promoter TIF can be scored independently and combined in a single model to predict full-length promoter TIF. Extending the models developed in this study with UP sequence predictions could create an opportunity to further increase the sequence flexibility to avoid unwanted sequence repetitions when engineering multiple operons. Further, Urtecho et al.[24] assessed how all individual promoter elements affect transcription and suggested that promoter background sequence considerably contributes to overall promoter expression through nonlinear interactions. Predicting how a promoter will perform regarding the larger genetic context, however, remains a tricky task. Besides the sequence of the promoter itself, e.g., DNA supercoiling, adjacent transcription factor binding sites, and competition between transcription factors impacts transcription. To eliminate potential unwanted adjacent genetic elements affecting transcription initiation and promoter escape, our promoter chassis was inserted in an insulator sequence spanning the −105 to +55 promoter region (previous work)[56,73]. Therefore, it is expected that the developed tool performs equally well for the transcription of different open reading frames. Also, by insulating the promoter region from nearby transcription factor binding sites, impact from environmental context, other than general changes in the expression of the cell's shared transcription machinery, is minimized. However,

changing the genetic context outside of the modeled region, and especially in the −105 to +55 region, would still jeopardize the robustness of any promoter TIF prediction model available.

In conclusion, a highly practical tool to predict the promoter TIF of σ70 promoters is developed and validated, as a valuable addition to the synthetic biology toolbox, providing the ability to engineer custom promoters and defined libraries. Based on the 95% confidence interval of the in vivo validation set, promoters can be reliably constructed spanning distinct ordinal catagories. This was enabled by both the construction of promoter libraries with randomized spacer sequences in *E. coli*, leading to data sets of an unprecedented size for this setting, and the recent developments in deep learning methods, which have been proven successful in topics involving big data. In addition, this tool can be further developed into a complete orthogonal expression toolbox by assembling the power of multiple prediction models for the TIF and orthogonality of multiple sigma factor specific promoters, boosting the potential of orthogonal genetic circuit design. Finally, the developed methodology will be useful in similar sequence-dependent ordinal problems, and the created data sets might contribute in revealing the essential spacer features contributing to transcriptional performance.

## Methods

**Media, strains, and plasmid construction**. All products were purchased from Sigma-Aldrich (Diegem, Belgium) unless otherwise stated. Agarose and ethidium bromide were purchased from Thermo Fisher Scientific (Erembodegem, Belgium). Standard molecular biology procedures were conducted as described by Sambrook et al.[74]. All DNA fragments were amplified using PrimeSTAR HS DNA polymerase (Takara, Westburg, Leusden, The Netherlands) and purified using the innuPREP PCRpure Kit (Analytik Jena AG, Jena, Germany).

Lysogeny broth (LB) was used for cloning purposes. Complex medium (853) was used for all further experiments. LB medium was composed of 10 g bacto-tryptone, 5 g yeast extract and 5 g NaCl in 1 L water. Eight hundred and fifty-three medium was composed of 10 g bacto-tryptone, 5 g yeast extract, 0.1% glucose, 5 g NaCl, 0.7 g $K_2HPO_4$ and 0.3 g $KH_2PO_4$ in 1 L water. Kanamycin (60 μg/mL) was added to all media for selection.

*E. coli* Top10 cells (Invitrogen, Carlsbad, USA) were used for cloning purposes. *E. coli* K12 MG1655 was used for all further experiments requiring promoter TIF measurements.

The construction of the promoter library expression vector (pLibrary), and its complete annotated DNA sequence, is described in detail in Bervoets et. al.[56], as well as the construction of all (heterologous) sigma factor specific promoter libraries.

The used σ70-specific promoter (library) sequences in pLibrary are derivatives of the insulated proD promoter, constructed by Davis et. al.[73]:

TTCTAGAGCACAGCTAACACCACGTCGTCCCTATCTGCTGCCC-TA**GGTCTATGAGTGGTTGCTGGATAACTTTACG** – 17 bp spacer – **TATAATATATTCA**GGGGAGAGCACAACGGTTTCCCTCTACAAA-TAATTTTGTTTAACTTT, with promoter sequence −60 to +1 in bold, and conserved −35 and −10 regions underlined. To insert forward engineered spacer sequences for the in vivo model validation, first a cloning vector was created, replacing the promoter region partially by an operon expressing the chromoprotein aeBlue[75] for visual selection (and later counterselection), flanked by restriction enzyme recognition sites for Golden Gate (GG) assembly[76]. Oligonucleotides and their reverse complements (IDT, Leuven, Belgium) of 60 basepairs, containing the engineered spacer sequences and GG restriction enzyme matching with the cloning vector, were duplexed as described by the manufacturer. The general duplexed DNA sequence is:

5′-*TTACG*GGG**TCTC**G*TTAC*G-17 bp spacer—TATAATAT*ATT***CAG**T*GAGACC*A*GCCA*-3′, with restriction enzyme recognition sites underlined and the restriction sites in bold. The DNA sequences surrounding the recognition sites (italic) are designed with a custom script using the ViennaRNA, RNAfold package[77] to minimize homodimer/monomer formation, to facilitate duplexing and the following GG assembly. All forward engineered 17 bp spacer sequences are given in Supplementary Table 2. Cloning vector containing aeBlue and subsequent forward engineered mKate2 expressing operons were sequence verified by Sanger sequencing (Macrogen Inc., Amsterdam, The Netherlands).

**Promoter TIF—sequence data acquisition**. The preparation of cells for fluorescence-activated cell sorting (FACS) is described in Bervoets et al.[56]. In short, a BD Influx Cell Sorter utilizing the BD FACS sortware sorter software (FACS core facility, cmpg, Leuven) and calibrated with Rainbow Calibration Particles (eight peaks, 3.0–3.4 μm) (BD), was used for cell sorting to split the promoter libraries in

presence of their cognate sigma factor into 12 separate bins, covering the entire expression range, to gather the genetic information of different promoter TIF levels. With a focus on the practical application, the ability to construct a promoter library with 12 levels of TIFs were deemed adequate. Additionally, further increase of the resolution becomes more obsolete as an effect of the normal distribution of the cellular fluorescence. Nonuniform bins are used to account for the variability in the distribution of the cells along the expression level. To account for the varying expression ranges of the bins, the predictive model was given an ordinal design. Also, the *B. subtilis* σB, σF, and σW-specific promoter libraries in presence of their noncognate sigma factors, were sorted in two bins to gather information about sequences resulting in conservation or loss of orthogonality. Constitutive sfGFP expression was taken into account for bin selection to exclude artifacts. A visual representation of the bin selection, the size of all population fractions and the number of sorted cells is given in Supplementary Figs. 1–4.

Sorted cells were collected in collection tubes with 1 mL 853 medium supplemented with kanamycin. Subsequently, sorted cells were added to 5 mL fresh 853 medium supplemented with kanamycin in 50 mL tubes and incubated overnight at 30 °C while shaking. Plasmid DNA was isolated for all individual bins with a Qiagen Plasmid Mini Kit (Qiagen, Venlo, The Netherlands) to prepare for DNA sequencing.

The followed sample preparation workflow for sequencing of the sorted libraries, is adapted from the "16S Metagenomic Sequencing Library Preparation" protocol (Illumina Inc.)[78]. The two-step PCR workflow creates amplicons of the region of interest in a first PCR reaction and includes a primer binding site for a second PCR reaction, adding sample-specific indexes and sequencer flow cell binding sites for subsequent sequencing.

Plasmid template concentrations were determined using a Quant-iT™ PicoGreen™ dsDNA Assay Kit (ThermoFisher Scientific). 0.7 ng template (≈maximum recommended template concentration) and 0.15 μM of each primer was added to 40 μL PrimeSTAR® HS (Takara Bio Inc.) PCR reactions to produce amplicons of 270–275 bp. The promoter region-specific primers are given in Supplementary Table 4. A DNA Clean & Concentrator Kit (Zymo Research) was used for DNA purification after the first PCR, and amplicons were analyzed on a LabChip GX (PerkinElmer Inc.). PCR conditions were optimized for each primer pair to ensure a DNA yield of minimum 1 ng/μL.

For index PCR reactions, a Nextera XT Index Kit (Illumina Inc.) and the KAPA HiFi HotStart ReadyMix (F. Hoffmann-La Roche Ltd) were used. Each of the 66 samples (12 bins × 4 libraries + 2 bins × 3 libraries × 3 different genetic backgrounds) was tagged with a unique pair of indexes. Subsequent DNA purification was performed using AMPure XP beads (Beckman Coulter Inc.), followed by LabChip analysis to determine amplicon concentrations. Samples were pooled together proportionally with the number of sorted cells for each bin (Supplementary Figs. 1–4). and taking into account the theoretical maximum amount of unique sequences for each library (= #transformants during library construction). To remove potential excess index primers and off-target amplicons, a DNA size-specific gel purification was performed (Zymoclean Gel DNA Recovery Kit, Zymo Research). Finally, quality control and sequencing (dual-index, single-read 50 bp sequencing, MiSeq, Illumina) was performed by the NXTGNT lab (Faculty of Pharmaceutical Sciences, UGent).

**Characterization of forward engineered promoters**. To characterize the forward engineered promoters, plasmid DNA was introduced in the cells on microtiter plate (MTP)-scale by adding 2 μL of DNA to 10 μL chemically competent cells prepared in TSS buffer[79,80] (5 g PEG 8000, 1.5 mL 1 M $MgCl_2$ and 2.5 mL DMSO supplemented with LB to 50 mL total) and a 45 s heat shock. Pre-cultures for analysis (eight biological replicates) (150 μL 853 medium with kanamycin) were grown for 24 h in sterile 96-well flat-bottomed black MTPs (Greiner Bio-One, Vilvoorde, Belgium), enclosed by a Breath-Easy® sealing membrane (Sigma-Aldrich), at 30 °C while shaking (800 rpm in a Compact Digital Microplate Shaker, ThermoFisher Scientific). Cultures were then diluted 300-fold (by serial dilution) in 150 μL fresh medium with kanamycin and cultured similarly to the precultures. Optical density at 600 nm ($OD_{600}$), mKate2 fluorescence (FL) (excitation: 588 nm, emission: 633 nm, gain: 115) and sfGFP FL (excitation: 480 nm, emission: 520 nm, gain: 80) were measured after reaching stationary phase in a Tecan Infinite m200 Pro plate reader (Tecan, Mechelen, Belgium). Management of the hardware and processing of the data is achieved using Tecan i-control and Magellan software. FL measurements were processed by first correcting mKate2 and sfGFP FL for growth media (blank) and subsequently, calculating the ratio of mKate2 over sfGFP FL ($Fluorescence_C$).

$$Fluorescence_C = \frac{FL(mKate2) - FL(mKate2)_{medium}}{FL(sfGFP) - FL(sfGFP)_{medium}}. \quad (1)$$

Reported values were obtained by a logarithmic transformation and normalization between 0 and 1, according to the following formula:

$$x = \frac{\log(P) - \log(P_{min})}{\log(P_{max}) - \log(P_{min})}, \quad (2)$$

with $x$ the transformed data and $P_{min}$ and $P_{max}$ the fluorescence_C value of the promoter displaying the lowest and highest TIF, respectively. Linear regression,

Spearman's rank correlation test and confidence and prediction intervals calculation is performed with a custom written R-script.

**Model building**. Data was processed using Python (NumPy, Pandas packages). Read samples containing mutations within the nonspacer region of the sequenced 51 bp were removed from the database, as well as sequence samples present in bin 0 (for sorted promoter libraries in presence of their cognate sigma factor), covering the sequences resulting in background levels of red FL. Several properties were described in order to better identify the relationship between a unique sequence and its distribution of reads $r$ throughout the observed bins. These are the total amount of reads present in all bins ($r_{tot} = \sum r_i$), the amount of local maxima $M$ with similar heights ($r_{bin_M - 1} < r_{bin_M} > r_{bin_M + 1}$ with $r_{bin_{M1}}/r_{bin_{M2}} > 0.66$ and $r_{bin_{M1}} > r_{bin_{M2}} > 0$), and the largest distance between the local maxima ($bin_{M1} - bin_{M2}$). Sequences containing outliers (exceeding 95th percentile) on given properties were removed from the data. The target label given to each sequence is defined as the bin with the highest count of reads.

The PyTorch library[81] was used for the purposes of building and using a shallow neural network model. Convolutions were applied in the first layers, having shown to be optimal for feature extraction (motifs) of DNA-protein interaction[62]. Several networks were evaluated before selecting the final model architecture. The structure and performances of several of the less complex networks are listed in Supplementary Table 5. The final network, schematically depicted in Figure 1b, first processes the sparse one-hot encoded input sequence by four sequential convolutions with 4 ($1 \times 1$), 16 ($1 \times 4$), and 32 ($1 \times 2$) kernels, respectively. The processed features are successively sent through a dropout layer ($p = 0.3$) and two fully connected layers with sizes 128 and 64.

In order to perform ordinal regression, for which a latent variable exists that is correlated to the promoter TIF represented by the eleven classes, a single linear combination of the output vector $\mathbf{x}$ with a set of weights $\mathbf{w}$ is taken. A final sigmoid transformation returns values between 0 and 1:

$$\hat{\mathbf{o}}_i = \text{Sigmoid}(\mathbf{wx} + \mathbf{b}_i), \tag{3}$$

where

$$\mathbf{o}_i(y) = \begin{cases} 0, & \text{if } i \geq y \\ 1, & \text{if } i < y \end{cases}, \qquad i \in [0, 9].$$

To ensure a prediction loss correlated to the distance of the predicted label with the ground truth, multilabel classification is performed. The sample loss equals

$$\zeta_{out} = \mathbf{w}_n \sum_{i=0}^{K} \text{BCE}(\hat{\mathbf{o}}_i, \mathbf{o}_i) = \mathbf{w}_n \sum_{i=0}^{K} [\mathbf{o}_i \cdot \log(\hat{\mathbf{o}}_i) + (1 - \mathbf{o}_i) \cdot \log(1 - \hat{\mathbf{o}}_i)], \tag{4}$$

where $K$ = number of classes.

The sum of the binary cross entropy (BCE) from the individual outputs thereby obtains the prediction loss $\zeta_{out}$. Importantly, a weight $\mathbf{w}_n$, having a negative correlation to the abundance of the sequence class label in the training data, is assigned to the loss of each sample, ensuring the importance of sequences from every class $n$ in an imbalanced setting.

To enforce the strict ordering of $\hat{\mathbf{o}}_i$, an auxiliary loss is calculated from the vector of biases $\mathbf{b}$ that serve as the delimiter of the classes. Specifically, the auxiliary training loss $\zeta_{aux}$ enforces the condition $\mathbf{b}_i > \mathbf{b}_{i+1}$ or $\hat{\mathbf{o}}_i > \hat{\mathbf{o}}_{i+1}$:

$$\zeta_{aux} = \sum_{i=0}^{K-1} \text{Softplus}(\mathbf{b}_{i+1} - \mathbf{b}_i) = \sum \ln\left(1 + e^{\mathbf{b}_{i+1} - \mathbf{b}_i}\right).$$

$$\zeta = \zeta_{out} + \zeta_{aux}. \tag{5}$$

the softplus function is a smoothed version of the rectified linear unit. This loss is minimal when the condition is met, albeit greater than zero to prevent oscillating behavior in the internal order of $\mathbf{b}$ during training. The total loss $\zeta$ of the network equals the sum of $\zeta_{out}$ and $\zeta_{aux}$.

Based on the target label and loss, the model is optimized for the output to approach $\hat{\mathbf{o}}_i = \text{Pr}(y > i | \mathbf{x})$. As such, we can deduce that a sample belongs to a given class as:

$$\text{Pr}(y = i | \mathbf{x}) = \text{Pr}(y > i | \mathbf{x}) - \text{Pr}(y > i + 1 | \mathbf{x}), \tag{6}$$

with

$$\text{Pr}(y > 10 | \mathbf{x}) = 0.$$

The architecture of the binary classification model predicting promoter orthogonality follows the same architecture, albeit without the specialized final layer and loss function. The training, validation and test data are created through randomized stratified sampling, ensuring equal class proportions for the three sets, and exist out of 70%, 10%, and 20% of the data, respectively. The model with a minimum in loss on the validation set is used for evaluation on the test set. For a given model architecture, multiple performance metrics on are obtained by evaluating five trained models using a 5-fold cross-validation scheme.

**Reporting summary**. Further information on research design is available in the Nature Research Reporting Summary linked to this article.

## Data availability

The data in this study is deposited on ArrayExpress (accession E-MTAB-8734). Source data are provided with this paper. All other relevant data are available from the authors upon reasonable request.

## Code availability

The proD tool and user guide is accessible to all and can be found at https://github.com/MEMO-group/ProD[82], with zenodo https://doi.org/10.5281/zenodo.4019340.

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

## Acknowledgements

The first author holds a PhD grant (141712) from the Institute for Innovation and Entrepreneurship in Flanders (Agentschap Innoveren & Ondernemen). This research was also supported by the BOF-IOP project "MLSB" (BOF16/ IOP/040) of the Bijzonder Onderzoeksfonds and by the BioRoboost Project of the European Union (820699).

## Author contributions

M.V.B., J.C., M.S., J.M., W.W., and M.D.M. were involved in the conception, design and writing of the manuscript. Wet lab experiments were performed by M.V.B., with a contribution by F.M. for cloning work. In silico work was performed by J.C. M.V.B. and J.C. performed the data analysis and interpretation of the results.

## Competing interests

The authors declare no competing interests.
