## [Peer Review File · Nature Communications]

Reviewers' Comments:

Reviewer #1:

Remarks to the Author:

"Predictive design of sigma factor-specific promoters", Maarten et. al.

The manuscript by Maarten and colleagues describes a method for the prediction of promoter strength (transcription initiation frequency) from sequence, which can be used in the forward design of promoter parts for synthetic biology. The approach takes advantage of convolutional neural networks, which is a machine learning approach that can fit complex non-linear functions. This is used to fit the relationship between the engineered spacer region and the promoter strength, ascertained from flow-sorting cells into eleven strength categories.

Overall, I think the paper is well written and logical. This paper builds on previous work, where the expression systems were introduced, and random promoter libraries generated and assayed. It seems the main contribution here is the model fitting and promoter strength prediction, which allows forward design of systems.

GENERAL COMMENTS

The authors present a CNN model of the sequences. In the conclusion, the authors highlight the power of CNN models. However, it is not shown that this approach outperforms a simpler method of regression. Given that this is a main aspect of the paper, I would like to see the performance comparison with a more standard linear regression/machine approach used in this area. One idea would be to fit a linear model containing the individual parameters and local interaction terms such as interaction terms between consecutive positions.

In Table 2 the standard errors of the AUC estimates are missing which makes it difficult to compare the values and whether they are statistically different.

I think the authors need to discuss more the fact that only promoter strength category can be (reasonably) predicted. This is important for people wanting to build reliable systems. Figure 3 shows highly overlapping predictions across categories and the caption states this "can at least partially be attributed to the expected biological noise". This must be expanded on in the discussion. Do the authors expect to understand and reduce this noise through statistics, model improvement, or better constructs?

The generality of the promoter expression systems is not discussed. Only the 17bp spacer region of the proD promoter is changed, which means that the majority of the promoter regions have the same sequence. Could this create problems with evolutionary stability when using multiple of these promoters in synthetic gene networks?

SPECIFIC COMMENTS

Pg18: PWN is not defined

Reviewer #2:

Remarks to the Author:

The manuscript by Maarten et al documents the design of promoter libraries for sigma factor. The paper builds several hundred thousand promoters and uses a shallow CNN to predict a strength-like metric. It is difficult to determine what the real contribution of the paper is: the authors themselves admit there is little to no biological insight (or generalisability) to be gained -see discussion on "context". In addition, the study is not novel in its nature: other works (Cuperus et

al, 2017) (de Boer et al, 2019) have explored similar problems with similar approaches. Notably: in the case of (de Boer et al, 2019), the *scale* of the study was *orders of magnitude* bigger than the one the authors report. In the case of (Cuperus et al, 2017) an attempt to explain what the artificial neural network learned and extract biological information was made with some success.

In addition, the rationale behind major choices of the paper is weak: why 12 categories? Why that specific gating strategy? How sensitive are the results to the choice of gating? The lack of a strong message, absence of innovation and the objectable choices all contribute to attenuate the intrinsic value of this contribution.

Additional points:

Introduction

- Line 48-49: brackets are not necessary
- Line 51: circuit, not circuitry
- Line 60: clarify "it's"
- Line 73 – 77: Sentence would benefit from being shortened to increase clarity
- Line 81: More up to date usage statistics?
- Line 54: "such a tool" an equivalent tool to the RBS calculator for promoter design
- Line 86: first step in gene expression.
- Line 109 – what is meant by highest predictive power? Reference?
- Line 114: Conserved regions are to be left unchanged to preserve orthogonality – is this completely necessary? Can a small number of changes be tolerated?
- Line 116: Not good writing to start a new paragraph with "this reason". What reason? E.g. to preserve promoter consensus regions...
- Line 116: don't need a comma after constructed or after nucleotides
- Line 118: define "wide range". An explanation of why you have built on the previous study in the way that you have done would be useful. E.g. was this study limited in terms of promoter library size? Activity levels? Predictability?
- Line 121: I am unclear as to what is meant by "per promoter". Does this mean you obtained x number of sequence variants in which the -35 and -10 regions were constant?
- Line 122: these data
- Line 124: What is the significance of 17 nucleotides?
- Line 125: Would benefit from a brief explanation of how you have "laid the in silico groundwork"

Results

- Line 137: The way this section is introduced makes it a little unclear on the first read as to what was done in the previous study and what work was done here.
- Line 138: what is the length of the spacer sequence? This is not stated in your introduction either
- Line 145: not sorting in accordance to transcription initiation frequency – sorting on the basis of cellular fluorescence, which is being used as a proxy.
- Line 145: why were 12 bins used? Arbitrarily chosen number? Looking at the supplementary figures, the bins are not equal in size. How was bin sizing defined?
- Line 148: remove binary
- Line 147: What is meant by "in the presence of non-cognate sigmas"? How was this achieved?
- Line 149: change wording of "similarly, ..." e.g. genotypic data were subsequently acquired through dna sequencing of the sorted promoter libraries.
- Line 153: how was the size of the buffer regions defined?
- Line 154 – 155: Are these percentages from all 4 libraries or just 1? Where on the promoter strength scale are the smallest bins found? e.g. weak or strong promoters? This is shown in supplementary info, but a brief descriptive sentence would be beneficial here.
- Line 154 – 155: Was a negative control used? E.g. WT cells, a promoter sequence that is known to have no in vivo activity? Do the sequences in the weakest expression bins actually have any significant in vivo promoter activity at all?
- Line 159: define "sizeable data set"

- Line 164: isn't it more accurate to say that multiple promoter libraries were created?
- Line 167: subsequently, not following
- Line 170: data sets?
- Line 180: why was in vivo validation only performed for the sigma 70 model?
- Line 187-189: a brief description of what the calculated data properties would be beneficial here. Related, in methods section line 563 you say sequences containing outliers on given properties were removed, but do not define what constituted an outlier.
- Line 189 – 190: The meaning of this sentence is unclear. Multiple sequences were shown to have read count distributions that were inconsistent...? A better definition of what you mean by "expected results" would be beneficial.
- Line 197: what is meant by better performance of the model? More accurate prediction when applied to test data?
- Line 202: training model based on classification to bin with most reads. But previously in text you acknowledge that promoter activity is inherently Gaussian in distribution. Is training the model in this way fully representative of promoter activity?
- Table 1: the labelling is very unclear. i.e. is original the total set size, filtered the portion used in model training? If so, the headings of the table should match the descriptions in the legend. If not, further description is required in the legend. In the left hand column, what does 0 and 1 refer to? Presence/absence of cognate sigma factor? Clarify. This would benefit from being visualised as a figure rather than represented as a table.
- Line 210: reference?
- Line 211: but doesn't all the preprocessing you've had to do show that this assumption is not valid? You've shown that promoters fall into different bins, have a Gaussian distribution of activity etc etc. now you're saying that you can assume that a single bin is representative of promoter function.
- Line 219: expand on what you mean by "can be tied to the architecture of the libraries"
- Line 229: reference?
- Figure 2: would benefit from labelling which shows how classes relate to promoter strength (i.e. does class one or class 10 contain the strongest promoters?). Scaling should also be made consistent – sigma 70 and sigma b plots are shorter than the other two plots. If possible, condensing the figure to fit on a single page would help. It might also be clearer if the class labelling were not added to every panel.
- Also, sigma F library suggests that G residue at the 5' end of the -10 becomes more prevalent as the promoter gets stronger. Could this difference be quantified?
- Line 250: Supplementary figure 5 is not included in the supplementary information. There are also errors in the legend where references are missing.
- Line 253 – 255: is this referring to a lack of identified nucleotides in spacer regions in the literature, or in your study? Requires clarification
- Line 261: some statistical analyses of the data to support this conclusion would be useful. E.g. statistical significance or otherwise of differences between groups in supplementary figure 6.
- Line 262: I think this statement needs to be tempered slightly – your data does not support the conclusions of the previous studies with regards to the four promoter classes you have studied, not more generally.
- Line 271: close bracket has no open bracket before it.
- Figure 3: not sure this is the best way to display these data? Not really sure what the take home message of the figure is meant to be. At the very least, needs more discussion in the text.
- Line 301: your model is not predicting TIF, it's classifying promoter sequences into expression bins
- Line 301: why did you decide to look at this class of promoters only for in vivo validation, rather than one of the other 3?
- Line 303: The nature of the monte carlo approach to promoter generation needs expanding on, either here or, probably more suitably, in the methods section.
- Line 308: training data set?
- Line 315: why are fluorescence values log transformed? A negative fluorescence value does not make biological sense.

- Line 333: should be rewritten to make it clearer that the promoter sequences that are being talked about here are taken from a previous study, not this investigation.
- Figure 4: might benefit from some discussion of misclassification – i.e. looking at the figure, some the activity levels of some of the promoters suggest that they do not belong to the bin to which they have been assigned by the model. How many sequences should have been assigned to different classes but were not?
- Line 331: some sort of visualisation of this data might be useful?

Discussion

- Line 345: a tool supporting de novo design of complete promoters is not shown in this study either. The results of the study should be placed in context – i.e. capable of predicting promoter activity based on variations in the spacer sequence in a single genetic context.
- Line 354: Clarify that the spacer DNA sequence is between the conserved -35 and -10 regions- there are other spacer sequences in promoters.
- Line 370: why is such a model non-sensible?
- Line 387: clarify what bin 0 is.
- Line 388: what were these reasons? Summarise here.
- Line 391: what was the nature of this filtering?
- Line 392: expand on what is meant by “the results”
- Line 400: needs more discussion of why PWMs are inadequate / references from literature
- An expanded discussion of the likely effects of genetic and environmental context on promoter activity would be beneficial. Your models perform well in the single context in which they have been developed and tested. How likely is it that model performance will hold when promoters are used to control transcription of a CDS other than the fluorescent protein used in their initial characterisation?
- Line 432: contextualise the claim on an “unprecedented size” data set – libraries cis-regulatory sequences of the size described here and larger have previously been described in the literature.

Methods

- Line 598: How were sequences assigned to training, validation and test sets?
- Line 540: what was the gain of the instrument?
- Line 566: typo, design
-

Supplementary material

- Supplementary table 3 caption: references “Chapter 3”. Replace with relevant reference.

Resubmission

Predictive design of sigma factor-specific promoters

Van Brempt Maarten^{1 †}, Jim Clauwaert^{2 †}, Friederike Mey¹, Michiel Stock², Jo Maertens¹, Willem Waegeman^{2 ‡} and Marjan De Mey^{1* ‡}.

¹ Centre for Synthetic Biology (CSB), Department of Biotechnology, Ghent University, 9000 Ghent, Belgium.

² KERMIT, Department of Data Analysis and Mathematical Modelling, Ghent University, 9000 Ghent, Belgium

* Corresponding author: Tel: +32 9 264 6028, Email: marjan.demey@ugent.be

† These authors contributed equally to the paper as first authors.

‡ These authors contributed equally to the paper as last authors.

Response to the reviewers

Note: references towards figures and tables are adjusted in the comments to the new paper layout.

Reviewer #1:

The manuscript by Maarten and colleagues describes a method for the prediction of promoter strength (transcription initiation frequency) from sequence, which can be used in the forward design of promoter parts for synthetic biology. The approach takes advantage of convolutional neural networks, which is a machine learning approach that can fit complex non-linear functions. This is used to fit the relationship between the engineered spacer region and the promoter strength, ascertained from flow-sorting cells into eleven strength categories.

Overall, I think the paper is well written and logical. This paper builds on previous work, where the expression systems were introduced, and random promoter libraries generated and assayed. It seems the main contribution here is the model fitting and promoter strength prediction, which allows forward design of systems.

Response: The authors thank the reviewer for the comment and feel encouraged by the overall positive comments and constructive remarks. We hope that the substantial changes we have made to the manuscript is sufficient to be accepted for publication.

1. The authors present a CNN model of the sequences. In the conclusion, the authors highlight the power of CNN models. However, it is not shown that this approach outperforms a simpler method of regression. Given that this is a main aspect of the paper, I would like to see the performance comparison with a more standard linear regression/machine approach used in this area. One idea would be to fit a linear model containing the individual parameters and local interaction terms such as interaction terms between consecutive positions.

Response: The authors thank the reviewer for the comment. The biggest advantage of neural networks is the ability to optimize a model on large data through gradient descent. Optimization through ordinary least squares (for linear regression) or application of other methods such as Support Vector Regression is not feasible given the size of the data.

Additionally, the implementation of ordinal regression is an important element to the study. Through manipulation of the network architecture and definition of a custom loss it is possible to optimize a model that penalizes the output prediction in line to the distance (in classes) of the true label. Machine learning approaches that inherently allow the support for ordinal regression are scarce.

These two properties of the prediction task are essential to the choice of applying a neural network for this setting. Nonetheless, evaluation of several model architectures has been done. Performances have not been included to the paper as it distracts from the main message, which is not centered on the architecture of the model, but on its application. In case it is wished for, these can be added to the article or the supplementary file. Four architectures are of importance to support the selection of the final architecture for the model (see Table R1 below). The first architecture (**A1**) is a single node, and is equal to linear regression. The second architecture (**A2**) is a fully connected layer. In theory, a single layer neural network with enough nodes can accurately represent any continuous function. In our setting, a fully connected layer with 64 nodes has been taken, which allows the mapping of local interaction terms. The third architecture (**A3**) is the implementation of two fully connected layers of 128 and 64 nodes, and is identical to the current network without the convolutional layers. The fourth architecture (**A4**) is the convolutional neural network. For all networks, the final layer forces ordinal regression, as explained in the paper. The same training, test and validation sets are used in each setting.

Table R1. Evaluation of four model architectures (A1: a single node, A2: a fully connected layer, A3: two fully connected layers of 128 and 64 nodes, and A4: convolutional neural network) for our four different sigma factor(σ)/promoter data sets to select the final model architecture. The best model architecture per parameter is given in bold. WLoss: weighted loss; WMAE: weighted mean absolute error.

	Model architecture	wLoss	WMAE	Accuracy	Pearson
σ^{70}	A1	0.081888359764966	1.82906539621296	0.189592999497907	0.507604435122851
	A2	0.08171525232167	1.82709630732489	0.192968981335191	0.508871134070336
	A3	0.077743435871481	1.65898108416407	0.216003600351582	0.563110575075169
	A4	0.075653556763657	1.62602507237522	0.230140724848454	0.572183743165092
σ^B	A1	0.121672439166332	1.83833765039183	0.195988385478643	0.532918971735711
	A2	0.12149275997444	1.84542728958669	0.196200779644329	0.533961533248259
	A3	0.114347615298223	1.66217275475397	0.229773913528387	0.540792785818849
	A4	0.11170647918603	1.63830371289069	0.23172590553251	0.56847440560332
σ^F	A1	0.051416067866218	2.12772816060023	0.14714799429522	0.457072586644342
	A2	0.051518678325049	2.13274233902392	0.148820803244184	0.455571344380904
	A3	0.047626000158117	1.92142472409498	0.226330795057233	0.490073438478504
	A4	0.047816054797207	1.90302785635945	0.228468539968336	0.504660063360554
σ^W	A1	0.030742919156296	2.50667878317849	0.128040010655227	0.249375831595625
	A2	0.030383284941677	2.48208140075308	0.125855851425612	0.26008955225991
	A3	0.030502365534693	2.4382853755169	0.133275450158229	0.254018923314117
	A4	0.030808066001813	2.53618285103798	0.121469099037247	0.21548997197538

In general, we can see that the added complexity of the (convolutional) layers with the given hyperparameters gave some improvement, most notably for the best performing data sets (σ^{70} , σ^B). For σ^W , the increased complexity of the model did not benefit the model performances. However, the bad performances of this data set is related to the quality of the data, and the decreased performances of more complex models are an effect of fitting noise. For these reasons, we have selected and used model architecture A3 in our research.

2. In Table 1 the standard errors of the AUC estimates are missing which makes it difficult to compare the values and whether they are statistically different.

Response: The authors thank the reviewer for the comment. The comment of the reviewer is not completely clear, and we are not sure how we can evaluate whether different AUC values are statistically different. It is possible to add error bars to the performance metrics, and this is what we will do for the revision.

However, we believe that it does not make sense to put p-values on those numbers. In general, there is a lot of debate in the machine learning community whether it makes sense to perform statistical tests to compare different learning algorithms. For example, applying a standard t-test or a non-parametric Wilcoxon test on the different folds obtained via cross-validation will violate standard statistical assumptions. The main problem is that the training sets overlap for the different folds, which makes that observations from different folds are not independent. This has been reported by numerous authors, for example (Dietterich et al., 1998, 'Approximate statistical tests for comparing classifiers', *Neural computation*). In this article, the author recommends the use of a five-by-two cross-validated t-test, which is known to be one of the most accurate procedures for evaluating machine learning algorithms. However, this procedure cannot be applied on deep learning methods, which require a lot of data for training, whereas the 5-by-2 cross-validated t-tests only uses 10% of the data for training.

In the newly submitted paper, we have left out the ROC AUC values for each class ($y= 0$ to 10) per sigma factor/promoter data set. These represented the ability of the model to separate between two sets of classes at all possible splits (e.g., $0 / 1,2,3,\dots$; $0,1 / 2,3,4,\dots$; ...). However, we felt this metric to be not as informative. Instead, mean absolute errors (MAE) for the samples of each class ($y= 0$ to 10) are given, the mean of which is equal to the weighted mean absolute error. This metric pinpoint the classes with the lowest/highest performance, and supports the interpretation of Figure 5.

The manuscript and Table 1 were modified accordingly.

Given the amount of data and the ordinal nature of the prediction task, a shallow convolutional neural network adapted for ordinal regression has been created and trained for each sigma factor/promoter data set (Figure 1B&C). See Materials and Methods for a full description of the model architecture and training. The model performances are given in Table 1. Weighted performance metrics are used to make up for the imbalance of sequence counts between the classes, thereby ensuring the importance of each class independent of the total amount of samples it contains. This equals to the unweighted results in the case of a balanced sample distribution. For each class separately, the mean absolute error is given, revealing the uncertainty of the model on class level. This is visually expanded upon in Figure 4, where the fraction of predicted labels for the samples of each class are given.

Table 1: Performance metrics of the models on the test sets. Weighted metrics are used for accuracy and mean absolute error to account for class imbalance. For each performance, the mean and standard deviation (Std.) are given obtained by training multiple models in a five-fold set-up for the test set. Mean absolute errors for each of the sample classes ($y=0$ to 10) are given. ROC/PR AUC is used for the binary classification problem. ROC AUC represent a perfect model at $AUC = 1$. (ACC: accuracy; MAE: mean absolute error; AUC: area under the curve; ROC: receiver operating characteristic; WT: wild-type; σ : sigma factor).

	σ^0 specific promoters		σ^B specific promoters		σ^F specific promoters		σ^W specific promoters	
	WT genotype		σ^B genotype		σ^F genotype		σ^W genotype	
	Mean	Std.	Mean	Std.	Mean	Std.	Mean	Std.
Spearman's rho	0.574	0.003	0.565	0.002	0.497	0.002	0.234	0.050
Weighted ACC	0.230	0.005	0.230	0.005	0.210	0.004	0.136	0.015
Weighted MAE	1.609	0.007	1.652	0.017	1.919	0.052	2.504	0.100
MAE								
$y = 0$	2.866	0.150	2.397	0.068	2.891	0.107	4.797	0.240
$y = 1$	1.845	0.105	1.630	0.070	2.097	0.117	4.022	0.141
$y = 2$	1.372	0.054	1.071	0.028	1.507	0.101	2.739	0.287
$y = 3$	1.284	0.058	1.197	0.035	1.294	0.056	1.613	0.329
$y = 4$	1.241	0.045	1.368	0.058	1.324	0.095	1.090	0.122
$y = 5$	1.324	0.023	1.407	0.055	1.357	0.049	0.587	0.480
$y = 6$	1.401	0.038	1.333	0.036	1.258	0.068	0.951	0.111
$y = 7$	1.362	0.045	1.315	0.051	1.572	0.119	1.790	0.189
$y = 8$	1.316	0.055	1.507	0.065	2.102	0.146	2.711	0.246
$y = 9$	1.533	0.032	1.730	0.116	4.010	0.135	3.774	0.205
$y = 10$	2.158	0.047	3.224	0.165	1.704	0.320	4.646	0.298
ROC AUC								
			σ^F genotype	σ^B genotype	σ^B genotype	σ^B genotype	σ^B genotype	σ^B genotype
			0.694	0.004	0.652	0.004	0.615	0.010
ROC AUC			σ^W genotype	σ^W genotype	σ^W genotype	σ^W genotype	σ^F genotype	σ^F genotype
			0.691	0.002	0.643	0.004	0.635	0.006
ROC AUC			WT genotype	WT genotype	WT genotype	WT genotype	WT genotype	WT genotype
			0.665	0.004	0.632	0.003	0.635	0.006

3. I think the authors need to discuss more the fact that only promoter strength category can be (reasonably) predicted. This is important for people wanting to build reliable systems. Figure 3 shows highly overlapping predictions across categories and the caption states this “can at least partially be attributed to the expected biological noise”. This must be expanded on in the discussion. Do the authors expect to understand and reduce this noise through statistics, model improvement, or better constructs?

Response: The authors thank the reviewer for the comment. We understand that this remark covers several related topics and have divided our comments accordingly.

On the subject of noise

Binning has been performed solely in correlation to the fluorescence of the expressed mKate2 protein, where bins are not delineated using normalized expression values. As an effect, both the extrinsic and intrinsic noise are present within the data sets on which the models are trained and evaluated. In contrast, the expression rates of the *in vivo* validation set have been adjusted for extrinsic noise (to a degree, e.g. by excluding effects of variance in the plasmid copy number, growth, etc.), and performance metrics derived thereof do therefore better represent the capability of the model as compared to those derived of the test set.

The distinction between factors of noise within the data sets has now been expanded upon in both the results and discussion section of the revised manuscripts.

On the subject of handling noise

No individual metrics of fluorescence is available for individual samples/promoter sequences, and statistical tests to detect or improve noise are therefore out of reach. From a machine learning point of view, handling of noise is done through regularization of the model to prevent overfitting. This is mainly achieved through implementation of the ordinal layer, which projects all the classes on a single latent

variable, albeit at the cost of a lower complexity of the output classes. Another factor is the use of a validation set for early stopping of the training procedure. The substantial difference between the spearman rank correlation of the test set and *in vivo* validation set proves that the model successfully regulates against noise.

The importance of regularization of the selected model is expanded upon in a large portion of the discussion of the revised manuscript.

On the subject of model capability

Based on the 95% confidence interval of the *in vivo* validation set, we determine that promoters can be reliably constructed distinct ordinal categories spanning consecutive classes. As the *in vivo* validation set has been adjusted for extrinsic noise, these better represent the capability of the model.

This aspect has been expanded upon in the last paragraph of the discussion of the revised manuscript.

4. The generality of the promoter expression systems is not discussed. Only the 17bp spacer region of the proD promoter is changed, which means that the majority of the promoter regions have the same sequence. Could this create problems with evolutionary stability when using multiple of these promoters in synthetic gene networks?

Response: The authors thank the reviewer for the comment. In our experience, working with multiple of these promoters does not result in instability issues as bacteria are less prone to instability issues caused by repetitive sequences in comparison to, for example, *Saccharomyces cerevisiae*. However, it might be expected that there is a limit to the amount of these promoters that can be used without problems. To this end, strains in which the native recombinase for homologous recombination (e.g. RecA in *E. coli*) is deleted could provide a solution. Additionally, introducing sequence flexibility in the promoter regions could also provide a solution. In the discussion of the revised manuscript, a perspective to increase sequence flexibility and avoid long repetitions by e.g., altering UP regions is proposed. The -10 and -35 conserved regions are the most critical parts for orthogonality, and consequently these cannot be changed. As it stands, these sequences are relatively short, and no instability issues are expected for increased repetitions.

Reviewer #2

1. The manuscript by Maarten et al documents the design of promoter libraries for sigma factor. The paper builds several hundred thousand promoters and uses a shallow CNN to predict a strength-like metric. It is difficult to determine what the real contribution of the paper is: the authors themselves admit there is little to no biological insight (or generalisability) to be gained -see discussion on "context". In addition, the study is not novel in its nature: other works (Cuperus et al, 2017) (de Boer et al, 2019) have explored similar problems with similar approaches. Notably: in the case of (de Boer et al, 2019), the *scale* of the study was *orders of magnitude* bigger than the one the authors report. In the case of (Cuperus et al, 2017) an attempt to explain what the artificial neural network learned and extract biological information was made with some success.

Response: The authors thank the reviewer for the comment. In this paper, we provide a tool for the forward engineering of promoter elements for the construction of genetic circuits in *E. coli*. This tool allows for a broad range of activities of sigma 70. For long over a decade multiple attempts have been made in providing such a tool and it is our conviction that this is the first prokaryotic promoter strength prediction and design tool with a real practical use for the bioengineering and synthetic biology community, significantly exceeding performance of previous work in both precision and sequence flexibility. In addition, the study offers additional tools for the construction of promoter elements in order to construct multiple, orthogonal, pathways. Orthogonal expression is a key characteristic in bioengineering and synthetic biology as it allows for no interference with the host's physiology and thus reduce undesirable interactions. Both of the objectives are *innovative, novel* and of high interest to the bioengineering and synthetic biology community, where the selection of promoter elements is, at this moment, still limited to existing (limited) libraries holding well-characterized promoter sequences (e.g. Anderson promoter collection).

In our research, the application of fluorescence activated cell sorting or neural networks (with the exception of the ordinal output layer) are previously described techniques and are not by themselves of importance towards the story of this paper. As such, concerns raised on the application of these techniques by other studies---featuring diverse objectives---are, to our understanding, not inhibitive to our contribution. More specifically, de Boer et. al. models the expression rate of a gene for a variable promoter sequence linked to a yeast transcription factor. In our study we have applied these techniques following a different objective. Besides, we have created an additional validation set to evaluate our model *in vivo* and made the model for the forward engineering of promoter sequences available to the public. We do not believe that an even larger data set would have influenced the results of our study further. Cuperus *et al.* on the other hand, focus on mapping the relationship between 5' UTR regions and the translation process in yeast. As noted, our study did not delve into the motifs learned by the artificial neural network. Primarily, this decision is founded on the limited insights motifs on the spacer sequence offer on the promoter strength. Given the use of artificial neural networks featuring multiple layers, we can expect motifs to be of limited capacity to represent higher order interactions. More importantly, as the spacer sequence has been mapped using a conserved promoter sequence (with exception of the randomized spacer), no guarantee can be given that any interpretation extracted from such a study can be generalized to any promoter sequence on the genome.

2. In addition, the rationale behind major choices of the paper is weak: why 12 categories? Why that specific gating strategy? How sensitive are the results to the choice of gating? The lack of a strong message, absence of innovation and the objectable choices all contribute to attenuate the intrinsic value of this contribution.

Response: The authors thank the reviewer for the comment. The selection of the number of sorting bins is in relation to the resolution (i.e. number of classes) one desires to classify transcription initiation frequency with. However, too many classes are redundant towards the creation of promoter libraries, might result in mostly overlapping expression ranges as an effect of extrinsic noise, and increase the cost of the study. As such, the number of optimal sorting bins is limited. In essence, the selection on the exact number of sorting bins and where they are bordered is a pseudo-arbitrary choice, not different from previous *Nature* publications (Mutalik et al, 2013; Cuperus et al, 2017; de Boer et al, 2019).

To account for the gating strategy, we have adapted the artificial neural network towards an ordinal prediction task, which addresses the concerns that come from having a prediction task with various sorting bins located at various distances from one another. The weighted loss addresses the concern of allocating equal importance to all classes, irrespective of their sample size. All contribute to attenuate the intrinsic value of this contribution, being the creation of a practical tool for designing *de novo* promoter libraries for different orthogonal expression systems.

To make this point clearer, the manuscript was modified accordingly:

With a focus on the practical application, the ability to construct a promoter library with 12 levels of TIFs were deemed adequate. Additionally, further increase of the resolution becomes more obsolete as an effect of the normal distribution of the cellular fluorescence. Nonuniform bins are used to account for the variability in the distribution of the cells along the expression level. To account for the varying sizes of the bins, the predictive model was given an ordinal design.

3. Additional points:

Introduction

- Line 48-49: brackets are not necessary
- Line 51: circuit, not circuitry
- Line 116: don't need a comma after constructed or after nucleotides
- Line 122: these data

Response: The authors thank the reviewer for the comments. We have corrected these typo's in the manuscript.

- Line 73 – 77: Sentence would benefit from being shortened to increase clarity
- Line 54: “such a tool” an equivalent tool to the RBS calculator for promoter design
- Line 86: first step in gene expression.
- Line 116: Not good writing to start a new paragraph with “this reason”. What reason? E.g. to preserve promoter consensus regions...
- Line 60: clarify “it's”

Response: The authors thank the reviewer for the comments. We have modified the sentences in the manuscript accordingly to increase clarity.

- Line 81: More up to date usage statistics?

Response: The authors thank the reviewer for the comment. The statistics have been updated.

- Line 109 – what is meant by highest predictive power? Reference?

Response: The authors thank the reviewer for the comment. We have adjusted the text to remove the incorrect statement.

- Line 114: Conserved regions are to be left unchanged to preserve orthogonality – is this completely necessary? Can a small number of changes be tolerated?

Response: The authors thank the reviewer for the comment. In principle a small number of changes can be tolerated, but loss of orthogonality is highly likely, and orthogonality is required between multiple sigma factor/promoter pairs. Changing 1 base pair might not result in the loss of orthogonality toward one sigma factor but increases the chance substantially to lose orthogonality toward at least one of the four studied sigma factors. To this end, all combinations should be tested again experimentally when a base pair is changed in one of the conserved regions.

To clarify this better, the manuscript was modified accordingly:

As the orthogonality of gene expression in synthetic genetic systems becomes increasingly important, the -35 and -10 conserved regions are to be left unchanged to ensure this property in the design of genetic circuits.

- Line 118: define “wide range”. An explanation of why you have built on the previous study in the way that you have done would be useful. E.g. was this study limited in terms of promoter library size? Activity levels? Predictability?

Response: The authors thank the reviewer for the comment. In our previous work we have demonstrated that the use of chimera RNA polymerases allows to create a functional and orthogonal expression system in *E. coli*. Furthermore, by randomizing the promoter spacer nucleotides spanning between the specific -35 and -10 conserved regions of each selected sigma factor/promoter pair allowed to diversify the promoter TIFs. For each of these promoter libraries, a set of about 10 promoters spanning the range of promoter TIFs was selected and characterized in dept. Here, we build upon this work by combining the use of fluorescence-activated cell sorting (FACS) on these promoter libraries with targeted high-throughput DNA sequencing to obtain considerably large data sets (250,000 to 400,000 unique sequences per setting) holding promoter sequence – function relationship information in view of building a bioengineering and synthetic biology tool to predict and forward engineer promoter TIFs.

The manuscript was modified accordingly:

In previous work, to preserve orthogonality of the promoter sequence with sigma factors, promoter libraries were constructed by randomizing the promoter spacer nucleotides spanning between the -35 and -10 conserved regions, resulting in a 5 log range of promoter TIFs. Here, we build upon this work by combining the use of fluorescence-activated cell sorting (FACS) on these libraries and targeted high-throughput DNA sequencing to obtain considerably large data sets (250,000 to 400,000 unique sequences per setting) holding promoter sequence – function relationship information in view of building a tool which can predict promoter TIFs and design promoter sequences with a specific TIF. A computational model was trained on these newly created data, to develop the first *in vivo* validated *Escherichia coli* (*E. coli*) σ^{70} Promoter TIF Designer tool, named ProD. Our tool is able to output the spacer sequence, constituting 17 variable consecutive nucleotides. Additionally, albeit lacking *in vivo* validation, predictive models for promoter strength and orthogonality have been trained and evaluated to expand such a tool with three different

promoter architectures with specificities toward heterologous *Bacillus subtilis* (*B. subtilis*) sigma factors B, F, and W.

- Line 121: I am unclear as to what is meant by “per promoter”. Does this mean you obtained x number of sequence variants in which the -35 and -10 regions were constant?

Response: The authors thank the reviewer for the comment. We have modified the sentences in the manuscript accordingly to increase clarity.

- Line 124: What is the significance of 17 nucleotides?

Response: The authors thank the reviewer for the comment. This refers to the sigma 70 spacer sequence. We have modified the sentences in the manuscript accordingly to increase clarity.

A computational model was trained on these newly created data, to develop the first in vivo validated *Escherichia coli* (*E. coli*) σ^{70} Promoter TIF Designer tool, named ProD. Our tool is able to output the promoter spacer sequence, constituting 17 variable consecutive nucleotides.

- Line 125: Would benefit from a brief explanation of how you have “laid the in silico groundwork”

Response: The authors thank the reviewer for the comment. We agree that a brief explanation of how we have “laid the *in silico* groundwork” would benefit the readability of the manuscript.

The manuscript was modified accordingly:

Additionally, albeit lacking *in vivo* validation, predictive models for promoter strength and orthogonality have been trained and evaluated to expand such a tool with three different promoter architectures with specificities toward heterologous *Bacillus subtilis* (*B. subtilis*) sigma factors B, F, and W.

- Line 148: remove binary
- Line 167: subsequently, not following
- Line 170: data sets?
- Line 250: Supplementary figure 5 is not included in the supplementary information. There are also errors in the legend where references are missing.
- Line 271: close bracket has no open bracket before it.

Response: The authors thank the reviewer for the comments. We have corrected these typo's in the manuscript.

- Line 145: not sorting in accordance to transcription initiation frequency – sorting on the basis of cellular fluorescence, which is being used as a proxy.
- Line 149: change wording of “similarly, ...” e.g. genotypic data were subsequently acquired through dna sequencing of the sorted promoter libraries.
- Line 159: define “sizeable data set”
- Line 164: isn't it more accurate to say that multiple promoter libraries were created?
- Line 187-189: a brief description of what the calculated data properties were would be beneficial here. Related, in methods section line 563 you say sequences containing outliers on given properties were removed, but do not define what constituted an outlier.
- Line 219: expand on what you mean by “can be tied to the architecture of the libraries”

- Line 301: your model is not predicting TIF, it's classifying promoter sequences into expression bins
- Line 308: training data set?
- Line 333: should be rewritten to make it clearer that the promoter sequences that are being talked about here are taken from a previous study, not this investigation.
- Line 387: clarify what bin 0 is.
- Line 388: what were these reasons? Summarise here.
- Line 392: expand on what is meant by "the results"
- Line 432: contextualise the claim on an "unprecedented size" data set – libraries cis-regulatory sequences of the size described here and larger have previously been described in the literature.
- Line 566: typo, design
- Supplementary table 3 caption: references "Chapter 3". Replace with relevant reference.

Response: The authors thank the reviewer for the comments. We have modified the sentences in the manuscript accordingly to increase clarity.

- Line 137: The way this section is introduced makes it a little unclear on the first read as to what was done in the previous study and what work was done here.

Response: The authors thank the reviewer for the comment. We agree with the reviewer and have modified the manuscript to better introduce the previous study and this study.

In previous work, we have demonstrated the use of chimera RNA polymerases based on heterologous sigma factors from *B. subtilis* that recognize specific promoter sequences to create a functional and orthogonal expression system in *E. coli*. To this end, the spacer sequence between the conserved -35 and -10 σ recognition sites of *E. coli* σ^{70} and *B. subtilis* σ^B , σ^F and σ^W -specific promoter sequences (17bp, 12bp, 15bp and 16bp, respectively) was engineered, to introduce variability in promoter TIF while preserving the orthogonal features toward specific σ s⁵⁶. A vector (pLibrary) was constructed for each σ -specific promoter, consisting of a promoter library site in a red fluorescent protein (mKate2⁶³) expressing operon, and a second operon, constitutively expressing sfGFP⁶⁴ as an internal reference for normalization (Figure 1A). In this work, to define the spacer sequence – function relationship, first, FACS was used to sort cells, in accordance to cellular fluorescence (proxying promoter TIF), into 12 sorting bins on all four promoter libraries. Next, high-throughput DNA sequencing of each bin was performed for promoter genotyping (Figure 1A). Additionally, the vectors containing the different promoter libraries were cloned into strains containing their non-cognate σ s in the genome. These cells were sorted into a non-fluorescent and fluorescent subpopulation, indicating conservation or loss of orthogonality, respectively. Similarly, through the use of high-throughput DNA sequencing, genotypic data was acquired.

- Line 138: what is the length of the spacer sequence? This is not stated in your introduction either

Response: The authors thank the reviewer for the comment. We agree with the reviewer that no clear indication of the specific spacer sequence length is given in the text (only in Figure 3) and have now included this in the manuscript.

The manuscript was modified accordingly:

To introduce variability in promoter TIF, while preserving the orthogonal features toward specific σ s, the spacer sequence between the conserved -35 and -10 σ recognition sites (17bp, 12bp, 15 bp and 16bp) was engineered in previous work, for *E. coli* σ^{70} and *B. subtilis* σ^B , σ^F and σ^W -specific promoter sequences, respectively⁵⁶.

- Line 145: why were 12 bins used? Arbitrarily chosen number? Looking at the supplementary figures, the bins are not equal in size. How was bin sizing defined?
- Line 153: how was the size of the buffer regions defined?

Response: The authors thank the reviewer for the comments. This has been expanded upon in the methods section. Please, see also comment 2.

The manuscript was modified accordingly:

With a focus on the practical application, the ability to construct a promoter library with 12 levels of TIFs were deemed adequate. Additionally, further increase of the resolution becomes more obsolete as an effect of the normal distribution of the cellular fluorescence. Nonuniform bins are used to account for the variability in the distribution of the cells along the expression level. To account for the varying sizes of the bins, the predictive model was given an ordinal design.

- Line 147: What is meant by “in the presence of non-cognate sigmas”? How was this achieved?

Response: The authors thank the reviewer for the comments. We have modified the sentences in the manuscript accordingly to increase clarity.

The manuscript was modified accordingly:

A vector (pLibrary) was constructed for each specific promoter, consisting of a promoter library site in a red fluorescent protein (mKate2 63) expressing operon, and a second operon, constitutively expressing sfGFP as an internal reference for normalization (Figure 1A). In this work, to define the spacer sequence – function relationship, first, FACS was used to sort cells, in accordance to cellular fluorescence (proxying promoter TIF), into 12 bins on all four promoter libraries. Next, high-throughput DNA sequencing of each bin was performed for promoter genotyping (Figure 1A). Additionally, the vectors containing the different promoter libraries were cloned into strains containing their non-cognate sigma factor in the genome.

- Line 154 – 155: Are these percentages from all 4 libraries or just 1? Where on the promoter strength scale are the smallest bins found? e.g. weak or strong promoters? This is shown in supplementary info, but a brief descriptive sentence would be beneficial here.

Response: The authors thank the reviewer for the comments. We have modified the sentences in the manuscript accordingly to increase clarity. In addition, we have visualized this in a new figure (Figure 2) replacing original Table 1.

The manuscript was modified accordingly:

The different bins account for a maximum of 20% to under 0.01% of the population size, ensuring the acquisition of genetic information of underrepresented promoter TIFs, such as seen for the lowest and highest rates (or bins) in all libraries (see Figure 2).

- Line 154 – 155: Was a negative control used? E.g. WT cells, a promoter sequence that is known to have no in vivo activity? Do the sequences in the weakest expression bins actually have any significant in vivo promoter activity at all?

Response: The authors thank the reviewer for the comments. In the previous work that this study builds on, the different promoter libraries were characterized, and the data was corrected with a negative control for visualization. However, for the purpose in this study, we believe using a negative control for cell sorting in the different expression bins is not a prerequisite for the further

workflow. First, the negative control on itself also shows as a gaussian distribution in flow cytometry, overlapping partly with the distribution of weak promoters. This makes it impossible to distinguish between the weakest promoters and sequences without *in vivo* activity in flow cytometry when analyzing libraries. Further, the created model uses ordinal classification, in which the lowest class could also be a class representing zero expression. Also, not being able to distinguish low from no activity results in the issue that cells containing mutations in the fluorescent protein gene or having other cell defects end up being sorted in the lowest bin, irrespective of their actual promoter activity. As a consequence, the lowest bin was not used for model training increasing the chance of sequences in the lowest class having significant *in vivo* promoter activity. Indeed, this is shown in the characterization of the forward engineered sequences having *in vivo* activity for class 0.

- Line 189 – 190: The meaning of this sentence is unclear. Multiple sequences were shown to have read count distributions that were inconsistent...? A better definition of what you mean by “expected results” would be beneficial.

Response: The authors thank the reviewer for the comments. We have modified the sentences in the manuscript accordingly to increase clarity.

The manuscript was modified accordingly:

A plethora of sequences have reads present in bins that provide incompatible properties, such as the presence of reads in distant bins constituting both low and high expression rates. Therefore, in order to exclude aberrations in the data, promoter sequences with outliers (5% highest values) on any of the data properties were removed.

- Line 197: what is meant by better performance of the model? More accurate prediction when applied to test data?

Response: The authors thank the reviewer for the comments. We have modified the sentences in the manuscript accordingly to increase clarity.

The manuscript was modified accordingly:

Absence of functional mKate2 expression might be caused by a variety of factors, such as mutations outside the promoter region or other cell defects. A preliminary analysis showed the exclusion of this bin to result in better accuracy of the model on the test set. This is not surprising, as the relative small sequence count and marginal position of the bin results in a high impact on the loss function of the ordinal regression setting, as a result of weighing the loss in function of the bin size (Material and Methods, 4.4.2).

- Line 202: training model based on classification to bin with most reads. But previously in text you acknowledge that promoter activity is inherently Gaussian in distribution. Is training the model in this way fully representative of promoter activity?

Response: The authors thank the reviewer for the comments. It might not be, but it is not feasible to define a prediction problem where the expression profile of the promoters is the output of the model. Within the article, we have expanded upon factors of biological noise (intrinsic, extrinsic) and argue these to influence the data of the FACS experiment. As we can not state whether the range of the expression profile is inherent to the promoter activity or the biological noise, denoting the bin with the highest reads as the label of the promoter strength is the safest choice. In practice--and other studies--the use of expression rates is by default the mean of the population.

- Table 1: the labelling is very unclear. i.e. is original the total set size, filtered the portion used in model training? If so, the headings of the table should match the descriptions in the legend. If not, further description is required in the legend. In the left hand column, what does 0 and 1 refer to? Presence/absence of cognate sigma factor? Clarify. This would benefit from being visualised as a figure rather than represented as a table.

Response: The authors thank the reviewer for the comments. We have modified the table into a figure (Figure 2) and have accordingly increased clarity.

- Line 210: reference?
- Line 229: reference?

Response: The authors thank the reviewer for the comments. We have included following references in the manuscript:

11. De Mey, M., Maertens, J., Lequeux, G. J., Soetaert, W. K. & Vandamme, E. J. Construction and model-based analysis of a promoter library for *E. coli*: an indispensable tool for metabolic engineering. *BMC Biotechnol.* 7, 34 (2007).
23. Jonsson, J., Norberg, T., Carlsson, L., Gustafsson, C. & Wold, S. Quantitative sequence-activity models (QSAM)—tools for sequence design. *Nucleic Acids Res.* 21, 733–739 (1993).

25. Jensen, K., Alper, H., Fischer, C. & Stephanopoulos, G. Identifying functionally important mutations from phenotypically diverse sequence data. *Appl. Environ. Microbiol.* 72, 3696–701 (2006).
26. Rhodius, V. A. & Mutalik, V. K. Predicting strength and function for promoters of the *Escherichia coli* alternative sigma factor, sigmaE. *Proc. Natl. Acad. Sci. U. S. A.* 107, 2854–9 (2010).
27. Meng, H. et al. Quantitative design of regulatory elements based on high-precision strength prediction using artificial neural network. *PLoS One* 8, e60288 (2013).
30. Brewster, R. C., Jones, D. L. & Phillips, R. Tuning promoter strength through RNA polymerase binding site design in *Escherichia coli*. *PLoS Comput. Biol.* 8, e1002811 (2012).
31. Meng, H., Ma, Y., Mai, G., Wang, Y. & Liu, C. Construction of precise support vector machine based models for predicting promoter strength. *Quant. Biol.* 5, 90–98 (2017).

- Line 211: but doesn't all the preprocessing you've had to do show that this assumption is not valid? You've shown that promoters fall into different bins, have a Gaussian distribution of activity etc etc. now you're saying that you can assume that a single bin is representative of promoter function.

Response: The authors thank the reviewer for the comments. We agree with the statement and have removed the wrong assumption

- Figure 3: would benefit from labelling which shows how classes relate to promoter strength (i.e. does class one or class 10 contain the strongest promoters?). Scaling should also be made consistent – sigma 70 and sigma b plots are shorter than the other two plots. If possible, condensing the figure to fit on a single page would help. It might also be clearer if the class labelling were not added to every panel.

Response: The authors thank the reviewer for the comments. The caption has been altered to relate the classes to promoter strength. The spacing between motifs is adapted to be equal between the two figures and class labeling is given a central position in between the motifs. The variability in the range of the different promoter types is due to the differences in spacer length.

- Also, sigma F library suggests that G residue at the 5' end of the -10 becomes more prevalent as the promoter gets stronger. Could this difference be quantified?

Response: The authors thank the reviewer for the comments. We have chosen to be cautious about the interpretation of given motifs. The creation of position weight matrices (PWM), albeit popular, solely represent first order interactions of the nucleotides' positions w.r.t. the fluorescence of the cells. PWMs are unable to represent any higher order interactions of the nucleotides and are therefore unable to represent correlations. PWMs are furthermore influenced by the presence of overrepresented patterns as an effect of the blueprints used (see line 219). The size of the G/T logo is, for example, attributed to the fact C/A nucleotides on that position have been excluded. As we chose to remain cautious on interpretation of given motifs, we are more so on the quantification of these.

- Line 253 – 255: is this referring to a lack of identified nucleotides in spacer regions in the literature, or in your study? Requires clarification

Response: The authors thank the reviewer for the comments. We have modified the sentences in the manuscript accordingly to increase clarity.

The manuscript was modified accordingly:

In contrast to the -35/-10 conserved regions and the UP-/extended -10-element, no studies have been published on the interactions of specific base pairs within the spacer with the cell's transcription machinery to this day. The contribution of spacer DNA to promoter TIF is most likely owed to structural features.

- Line 261: some statistical analyses of the data to support this conclusion would be useful. E.g. statistical significance or otherwise of differences between groups in supplementary figure 6.

Response: The authors thank the reviewer for the comments. The spearman rank correlations, used to evaluate a linear relationship between the GC-content and the promoter strength, is used as a non-parametric test and is necessary given the ordinal division of the samples. For each promoter we obtain:

F_F: correlation= -0.09234831427409075, pvalue=0.0

B_B: correlation= -0.17917085140101294, pvalue=0.0

W_W: correlation= 0.04464728049065141, pvalue=4.798286979631297e-130

RPOD: correlation= 0.05063223909087513, pvalue=5.03299935427998e-219

Showing RPOD and W_W to not be negatively correlated on a macro level.

However, as a result of the large sample sizes in each bin, the hypothesis that the sequences within two bins are sampled from the same distribution is almost always rejected using the non-parametric Mann–Whitney U test (see appendix of this rebuttal). Thus, there is no linear correlation between the GC-content and the strength, as the means between the bins are not (strictly) ordered. This is in contrast to the overall correlation, as given by the spearman test

This information has been added to the caption of Supplementary Figure 6.

Supplementary Figure 6. GC content (mean and standard deviation) of the promoter spacer libraries for each set of sequences assigned to a specific bin. Data is given for the Escherichia coli sigma factor 70 (σ^{70}) and Bacillus subtilis σ^B , σ^F and σ^W -specific promoter libraries, sorted in presence of their cognate σ . (WT: wild-type). The spearman rank coefficient over all the bins for each promoter is 0.05 for σ^{70} , -0.18 for σ^B , -0.09 for σ^F and 0.04 for σ^W ($p < 10^{-100}$). However, the Mann–Whitney U rejects the hypothesis that consecutive bins are sampled from the same distributions (data not shown, exceptions are bin 3-4 for σ^{70} , bin 0-1 for σ^B , bin 2-3 for σ^F and bin 9-10 for σ^W). As the means are not strictly ordered, GC-content of the promoters do not follow a linear relationship.

- Line 262: I think this statement needs to be tempered slightly – your data does not support the conclusions of the previous studies with regards to the four promoter classes you have studied, not more generally.

Response: The authors thank the reviewer for the comments. We agree with the reviewer.

The manuscript was modified accordingly:

With no linear correlations found, these previous findings are not supported for the four studied promoter chassis.

- Figure 4: not sure this is the best way to display these data? Not really sure what the take home message of the figure is meant to be. At the very least, needs more discussion in the text.

Response: The authors thank the reviewer for the comments. A clear ordinal correlation can be observed from the false positive predictions, as these are centered around the true label. This highlights the advantage of the model architecture. Section 2.4 and the caption of Figure 4 has been significantly changed to better expand upon the visualized results.

- Line 301: why did you decide to look at this class of promoters only for *in vivo* validation, rather than one of the other 3?
- Line 180: why was *in vivo* validation only performed for the sigma 70 model?

Response: The authors thank the reviewer for the comments. With a limited amount of resources for the study, the *in vivo* validation was only performed on the sigma 70 promoter. Although no *in vivo* validation has been performed on all library sets, the validation for sigma 70 proves the experimental set-up of the study. Sigma 70 was chosen over the others as it is of higher interest for the scientific community. Additionally, a limited validation is provided for the models for orthogonal predictions using the selected promoters of the previous study. This is depicted in an additional figure (Figure 6).

The manuscript was modified accordingly to clarify this better:

The model that is trained to predict TIFs of sigma 70-specific promoters in ordinal classes was subsequently subjected to *in vivo* validation. The sigma 70 model was selected due to its relevance for the scientific community.

...

Additionally, as a preliminary indication of the *in vivo* performance of model predictions for σ^B , σ^F and σ^W -specific promoters, we predicted the TIF class and orthogonality of a limited set of library promoters that were constructed and characterized in our previous study⁵⁶. The tested promoters do show an ordinal relation between measured expression level and predicted class, especially for the σ^B -specific promoters, though the number of observations is small and a fraction of the sequences was also present in the model training sets (Figure 6).

- Line 303: The nature of the monte carlo approach to promoter generation needs expanding on, either here or, probably more suitably, in the methods section.

Response: The authors thank the reviewer for the comments. The text has been altered to denote the use of randomly generated sequences. We agree that this is more proper compared to referring to a broad set of techniques.

- Line 315: why are fluorescence values log transformed? A negative fluorescence value does not make biological sense.

Response: The authors thank the reviewer for the comments. Log transforming fluorescence values is standard practice because the magnitude in differences in cellular response generated by the difference in expression is better described. Normalization is changed to values between 0 and 1 to make biological sense.

- Figure 5: might benefit from some discussion of misclassification – i.e. looking at the figure, some the activity levels of some of the promoters suggest that they do not belong to the bin to which they have been assigned by the model. How many sequences should have been assigned to different classes but were not?

Response: The authors thank the reviewer for the comments. Most importantly, it is not possible to translate the measured fluorescence to predicted outputs as the boundaries of each class are not delimited on the scale of fluorescence, making selection of predictions ‘outside of their respective bin’ unrealistic. Several promoters indeed show activity that is higher than the activity of

their neighboring classes (e.g. 4 and 7). As a degree of overlap is expected due to the intrinsic variance, we believe that it is furthermore unrealistic to define and select the outliers given the sample sizes (e.g. class 8,9 and 10 show variance of different degrees but are nicely centered). In contrast, we have added the Spearman correlation, Rsquared and the confidence and prediction intervals as these are statistically-substantial metrics.

- Line 331: some sort of visualisation of this data might be useful?

Response: The authors thank the reviewer for the comments. A new figure has been added to the revised manuscript depicting the discussed results (Figure 6).

- Line 345: a tool supporting de novo design of complete promoters is not shown in this study either. The results of the study should be placed in context – i.e. capable of predicting promoter activity based on variations in the spacer sequence in a single genetic context.

Response: The authors thank the reviewer for the comments. We agree with this comment. It is pointed out several times within the article that the given study has developed a tool for the *de novo* engineering of sigma-specific promoters that are bound by a given -10/-35 box. We do not believe this sentence implies we have achieved this.

- Line 354: Clarify that the spacer DNA sequence is between the conserved -35 and -10 regions- there are other spacer sequences in promoters.

Response: The authors thank the reviewer for the comments. In view of clarity, correct use of biology parts and standardization in synthetic biology, we opt here to make a clear separation between the different control elements and the term “core promoter” in the manuscript demarks the sequence region of the promoter sensu stricto without up elements or down elements. We

agree that the description of these different control elements is not always clear and have included a better definition and description of a core promoter in the manuscript.

The manuscript was modified accordingly:

The promoter libraries were designed by randomizing only the spacer DNA sequence delimited by the conserved -35 and -10 regions, leaving the conserved -35 and -10 regions unaltered to preserve specific sigma factor recognition, an essential feature in orthogonal genetic circuits⁵⁶.

- Line 370: why is such a model non-sensible?

Response: The authors thank the reviewer for the comments. We have modified the sentences in the manuscript accordingly to increase clarity.

The manuscript was modified accordingly:

However, due to the high capacity of neural networks and the aforementioned noise present within the data, predictions of given models can result in non-sensible and impractical predictions, such as similar probabilities for distant classes and dissimilar probabilities for bordering ones.

- Line 391: what was the nature of this filtering?

Response: The authors thank the reviewer for the comment. We have rephrased the sentence. The filtering denoted the fact that bin 0 should not cover the full lower bound of mKate2 expression rate, thereby excluding aberrant cells.

The manuscript was modified accordingly:

However, this bin is essential in solving the binary classification problem. We made observations indicating that models for prediction of orthogonality could be enhanced by improving the selection of the lower bins, ensuring the exclusion of aberrant cells, but more in depth research is required to assess the potential reduction of robustness of the model performances.

- Line 400: needs more discussion of why PWMs are inadequate / references from literature

Response: The authors thank the reviewer for the comment. We have added the required reference.

- An expanded discussion of the likely effects of genetic and environmental context on promoter activity would be beneficial. Your models perform well in the single context in which they have been developed and tested. How likely is it that model performance will hold when promoters are used to control transcription of a CDS other than the fluorescent protein used in their initial characterisation?

Response: The authors thank the reviewer for the comment. The authors agree that genetic and environmental context impacts transcription. This is discussed for UP- and background-sequence in the discussion already. To be more complete, the discussion is expanded for influence on different open reading frames and environmental context.

The manuscript was modified accordingly:

To eliminate potential unwanted adjacent genetic elements affecting transcription initiation and promoter escape, our promoter chassis was inserted in an insulator sequence spanning the -105 to +55 promoter region (previous work)^{56,72}. Therefore, it is expected that the developed tool performs equally well for the transcription of different open reading frames. Also, by insulating the promoter region from nearby transcription factor binding sites, impact from environmental context, other than general changes in the expression of the cell's shared transcription machinery, is minimized. However, changing the genetic context outside of the modeled region, and especially in the -105 to +55 region, would still jeopardize the robustness of any promoter TIF prediction model available.

Methods

- Line 598: How were sequences assigned to training, validation and test sets?

Response: The authors thank the reviewer for the comment. We have clarified this in the manuscript in the method section.

The training, validation and test data are created through randomized stratified sampling, ensuring equal class proportions for the three sets, and exist out of 70%, 10% and 20% of the data, respectively.

- Line 540: what was the gain of the instrument?

Response: The authors thank the reviewer for the comment. We have no included the gain value in the Material and Methods section:

Optical density at 600 nm (OD_{600}), mKate2 fluorescence (FL) (excitation: 588 nm, emission: 633 nm, gain: 115) and sfGFP FL (excitation: 480 nm, emission: 520 nm, gain: 80) were measured after reaching stationary phase in a Tecan Infinite m200 Pro plate reader (Tecan, Mechelen, Belgium).

Additional adjustments

Although the paper features many changes of that can not all be address in the rebuttal, several major changes should be listed.

The information from (what used to be) Table 1 has been transformed into a figure. To improve processing of the data for each setting, the information present in Table 1 is depicted in a new figure (now Figure 2).

Table 1: PR AUC values have been removed. The PR AUC--and any other metrics correlated to the relative class distributions--is only informative when the test set it is evaluated on reflects the ratio of positive and negative samples of the population the model is applied to in a practical setting. For example, consider training and evaluating a model that detects a virus from blood samples where an equal amount of positive and negative samples is present. The trade-off between the precision and recall of the obtained PR curve by themselves do not lend much insight on the performance of the model when applied on a population where a much lower occurrence of the virus is expected. Similar to this study, the class sizes of the binary classification to predict orthogonality are not reflective of the true population. In similar manner, this is why the weighted accuracy and weighted mean absolute error are used for the ordinal regression setting.

Table 1: MAE scores are given for each class for the TIF prediction problems. To better understand the performance of the model for the individual classes of each prediction task, the MAE is given for each class.

Appendix:

Table: log10 p-values on the GC percentage between sample bins using the Mann–Whitney U test. $\log_{10}(0.05) = -1.3$

ROPD												
Log10(p)	Bin 0	Bin 1	Bin 2	Bin 3	Bin 4	Bin 5	Bin 6	Bin 7	Bin 8	Bin 9	Bin 10	Bin 11
Bin 0		-6.28	-3.31	-4.12	-0.78	-16.14	-23.45	-26.77	-19.04	-13.5	-6.62	-11.82
Bin 1			-2.12	-2.19	-13.91	-48.5	-60.01	-64.68	-51.43	-41.64	-24.29	-31.73
Bin 2				-0.37	-10.83	-62.55	-82.14	-89.69	-66.12	-49.01	-22.04	-30.28
Bin 3					-18.79	-112.74	-149.89	-162.37	-113.46	-79.21	-28.63	-37.67
Bin 4						-86.34	-141.43	-159.21	-86.71	-46.33	-10.25	-18.33
Bin 5							-9.13	-16.76	-3.29	-0.48	-1.94	-1.02
Bin 6								-2.43	-1.26	-5.72	-5.57	-0.6
Bin 7									-4	-9.98	-7.72	-1.2
Bin 8										-2.63	-3.7	-0.32
Bin 9											-1.45	-1.15
Bin 10												-2.25
B_B												
Log10(p)	Bin 0	Bin 1	Bin 2	Bin 3	Bin 4	Bin 5	Bin 6	Bin 7	Bin 8	Bin 9	Bin 10	Bin 11
Bin 0		-0.48	-2.96	-6.68	-91.62	-300.9	-275.84	-inf	-inf	-inf	-299.53	-68.74
Bin 1			-2.93	-6.46	-67.1	-227.66	-243.3	-inf	-inf	-inf	-272.14	-67.4
Bin 2				-2.96	-89.31	-inf	-279.08	-inf	-inf	-inf	-302.49	-64.18
Bin 3					-24.43	-138.4	-179.21	-277.98	-283.48	-254.17	-213.67	-49.39

Bin 4						-110.05	-143.58	-256.4	-257.8	-221.55	-179.31	-27.65
Bin 5							-29.63	-81.72	-96.67	-92.51	-59.57	-3.15
Bin 6								-8.62	-16.53	-21.21	-7.3	-3.2
Bin 7									-2.76	-5.79	-0.32	-12.37
Bin 8										-1.57	-2.31	-18.7
Bin 9											-4.93	-22.91
Bin 10												-11.58
F_F												
Log10(p)	Bin 0	Bin 1	Bin 2	Bin 3	Bin 4	Bin 5	Bin 6	Bin 7	Bin 8	Bin 9	Bin 10	Bin 11
Bin 0		-1.61	-3.16	-32.35	-3.24	-116.37	-159.32	-297.14	-259.97	-153.95	-42.26	-5.03
Bin 1			-0.69	-15.68	-0.59	-102.69	-150.74	-276.1	-244.59	-148.92	-42.95	-3.71
Bin 2				-15.81	-0.41	-143.58	-169.01	-297.79	-258.12	-149.78	-44.66	-2.99
Bin 3					-20.11	-303.9	-244.01	-inf	-inf	-173.22	-62.36	-0.78
Bin 4						-172.54	-178.55	-inf	-266.63	-150.99	-45.05	-2.93
Bin 5							-44.45	-144.88	-132.7	-93.37	-10.65	-19.73
Bin 6								-27.12	-32.27	-43.81	-0.53	-45.83
Bin 7									-1.61	-14.03	-13.59	-84.81
Bin 8										-9.48	-17.22	-90
Bin 9											-31.45	-102.68
Bin 10												-33.81
W_W												
Log10(p)	Bin 0	Bin 1	Bin 2	Bin 3	Bin 4	Bin 5	Bin 6	Bin 7	Bin 8	Bin 9	Bin 10	Bin 11
Bin 0		-1.59	-5.91	-0.53	-134.14	-inf	-133.18	-43.21	-18.55	-5.03	-5.46	-0.34
Bin 1			-1.61	-0.88	-80.28	-172.27	-47.87	-21.89	-11.84	-5.17	-6.04	-1.13
Bin 2				-3.13	-108.79	-237.47	-76.65	-38.92	-23.77	-12.88	-12.73	-3.37
Bin 3					-73.11	-160.97	-40.59	-17.01	-8.3	-2.93	-3.94	-0.5
Bin 4						-8.31	-30.61	-57.89	-80.06	-107.95	-61.8	-55.71
Bin 5							-296.03	-inf	-inf	-inf	-175.51	-113.46
Bin 6								-22.67	-51.96	-96.77	-28.2	-27.26
Bin 7									-6.36	-22.41	-6.59	-11.25
Bin 8										-6.07	-1.37	-5.32
Bin 9											-1.18	-1.76
Bin 10												-2.64

Reviewers' Comments:

Reviewer #1:

Remarks to the Author:

"Predictive design of sigma factor-specific promoters", Maarten et. al.

Generally speaking I'm happy with the changes to the manuscript and the response to the comments and suggestions.

Regarding the model comparison. I think this is a useful demonstration of the benefit of the additional complexity of the model and convolutional layers. I think the authors made a mistake in the rebuttal as I believe they went ahead with model A4 not A3. (I don't agree with the statement that neural networks are the only approach that can take large data sets, as stochastic gradient descent is a general method that can be widely applied).

Although the authors state that this model comparison distracts from the main story, I think that it is of interest to other researchers in the field undertaking these kinds of machine learning approaches to predict the effect of sequence on phenotype. As such, I think it should be included in the supplementary information.

Reviewer #2:

Remarks to the Author:

The authors undoubtedly made significant efforts to improve the manuscript. Crucially, however, the key concerns I raised, remain unaddressed.

In particular, the generalisability of the developed tool was and remains limited - the authors present a tool that allows the modelling and design of synthetic sigma 70 promoters, with minimal evidence of applicability beyond that context. There is no element in the authors rebuttal to persuade that the paper advances the state-of-the-art in the application of CNNs for the design of regulatory sequences.

Also, in their rebuttal the authors state that "this is the first prokaryotic promoter strength prediction and design tool with real practical use". This is factually incorrect: there are other papers (including some by this same group) that try and address the same problem using other statistical modelling methods, albeit with admittedly limited validation of forward design capability. The work presented in this paper is a somewhat novel application of CNNs to the stated problem, but as discussed before it doesn't really advance the state-of-the-art, and lacks an attempt to extract biologically relevant information a la Cuperus etc. I can understand the authors' reluctance to try and perform this sort of analysis (second paragraph of the rebuttal to point 1, page 6), but this was and remains a key limitation of this study.

What the authors have developed (and they state this themselves in their rebuttal letter, page 17) is a tool for the engineering of promoters that are bound by a given -35/-10 box: a commendable work that, crucially, has little to add to what was known before the authors endeavoured in this effort.

Resubmission

Predictive design of sigma factor-specific promoters

Van Brempt Maarten^{1 †}, Jim Clauwaert^{2 †}, Friederike Mey¹, Michiel Stock², Jo Maertens¹, Willem Waegeman^{2 ‡} and Marjan De Mey^{1* ‡}.

¹ Centre for Synthetic Biology (CSB), Department of Biotechnology, Ghent University, 9000 Ghent, Belgium.

² KERMIT, Department of Data Analysis and Mathematical Modelling, Ghent University, 9000 Ghent, Belgium

* Corresponding author: Tel: +32 9 264 6028, Email: marjan.demey@ugent.be

† These authors contributed equally.

‡ These authors jointly supervised this work.

Response to the reviewers

Note: references towards figures and tables are adjusted in the comments to the new paper layout.

Reviewer #1:

Generally speaking I'm happy with the changes to the manuscript and the response to the comments and suggestions.

Regarding the model comparison. I think this is a useful demonstration of the benefit of the additional complexity of the model and convolutional layers. I think the authors made a mistake in the rebuttal as I believe they went ahead with model A4 not A3. (I don't agree with the statement that neural networks are the only approach that can take large data sets, as stochastic gradient descent is a general method that can be widely applied).

Although the authors state that this model comparison distracts from the main story, I think that it is of interest to other researchers in the field undertaking these kinds of machine learning approaches to predict the effect of sequence on phenotype. As such, I think it should be included in the supplementary information.

Response: The authors thank the reviewer for the comment. We apologize for the mistake in the rebuttal. Indeed, we went ahead with model A4. We agree that neural networks are not the only approach that can take large data sets; it was not our intention to insinuate this. We opt here for neural networks as they allow for an easy implementation of ordinal regression, the application of convolutional layers, which have shown to be successful in processing nucleotide sequences before, and the capability to process large data sets.

We agree with the reviewer that this model comparison could be of interest to other researchers in the field and have added the table comparing the different models to the supplementary information. The manuscript is modified accordingly:

Manuscript (start of Section 4.4.2)

The PyTorch library⁸⁰ was used for the purposes of building and using a shallow neural network model. Convolutions were applied in the first layers, having shown to be optimal for feature extraction (motifs) of DNA-protein interaction⁸¹. **Several networks were evaluated before selecting the final model architecture. The structure and performances of several of the less complex networks are listed in Supplementary Table 5.** The final network, schematically depicted in Figure1B, first processes the sparse one-hot encoded input sequence by four sequential convolutions with 4 (1 × 1), 16 (1 × 4), and 32 (1 × 2) kernels, respectively. The processed features are successively sent through a dropout layer ($p=0.3$) and two fully connected layers with sizes 128 and 64.

Supplementary Files

Supplementary Table 5. Performances of four of the evaluated model architectures for each of the four different data sets (sigma factor 70 (σ^{70}), σ^B , σ^F and σ^W -specific promoters) in order to select the optimal model architecture. The architectures were selected as they represent varying levels of complexity. The four architectures are a single node with no activation function, equaling linear regression (A1), a fully connected layer (A2), two fully connected layers of 128 and 64 nodes (A3), and the convolutional neural network used in the paper (A4). Every architecture has the ordinal layer as explained in the work. The best model architecture per metric is given in bold. More complex models did not yield better results. (WL: weighted loss; WMAE: weighted mean absolute error). Source data are provided as a Source Data file.

Data set	Model architecture	WL	WMAE	Accuracy	Spearman's rho
σ^{70}	A1	0.082	1.829	0.190	0.508
	A2	0.082	1.827	0.193	0.509
	A3	0.078	1.659	0.216	0.563
	A4	0.076	1.626	0.230	0.572
σ^B	A1	0.122	1.838	0.196	0.533
	A2	0.121	1.845	0.196	0.534
	A3	0.114	1.662	0.230	0.541
	A4	0.112	1.638	0.232	0.568
σ^F	A1	0.051	2.128	0.147	0.457
	A2	0.052	2.133	0.149	0.456
	A3	0.048	1.921	0.226	0.490
	A4	0.048	1.903	0.228	0.505
σ^W	A1	0.031	2.507	0.128	0.249
	A2	0.030	2.482	0.126	0.260
	A3	0.031	2.438	0.133	0.254
	A4	0.031	2.536	0.121	0.215

Reviewer #2:

The authors undoubtedly made significant efforts to improve the manuscript. Crucially, however, the key concerns I raised, remain unaddressed.

In particular, the generalisability of the developed tool was and remains limited - the authors present a tool that allows the modelling and design of synthetic sigma 70 promoters, with minimal evidence of applicability beyond that context. There is no element in the authors rebuttal to persuade that the paper advances the state-of-the-art in the application of CNNs for the design of regulatory sequences.

Also, in their rebuttal the authors state that "this is the first prokaryotic promoter strength prediction and design tool with real practical use". This is factually incorrect: there are other papers (including some by this same group) that try and address the same problem using other statistical modelling methods, albeit with admittedly limited validation of forward design capability. The work presented in this paper is a somewhat novel application of CNNs to the stated problem, but as discussed before it doesn't really advance the state-of-the-art, and lacks an attempt to extract biologically relevant information a la Cuperus etc. I can understand the authors' reluctance to try and perform this sort of analysis (second paragraph of the rebuttal to point 1, page 6), but this was and remains a key limitation of this study.

What the authors have developed (and they state this themselves in their rebuttal letter, page 17) is a tool for the engineering of promoters that are bound by a given -35/-10 box: a commendable work that, crucially, has little to add to what was known before the authors endeavoured in this effort.

Response: The authors thank the reviewer for the comment.

We don't agree with the statement of the reviewer that the paper doesn't really advance the state-of-the-art. In this paper, we provide a tool for the forward engineering of promoter elements for the construction of genetic circuits in *E. coli*. This tool allows for a broad range of activities of sigma 70 cognate promoters. We agree that other papers (including some of our own papers) have tried to create models that can predict promoter strength, however, with limited success (based on the results on the validation set). The model described in this paper displayed a robust performance for predicting promoter strength given a DNA sequence, but also allows for forward engineering of novel DNA sequences with a specific desired promoter strength. This model is also made available as a design tool (10.5281/zenodo.4019340) with a real practical use for the bioengineering and synthetic biology community, significantly exceeding performance of previous work in both precision and sequence flexibility.

In addition, this tool is expanded to the prediction and forward engineering of three additional classes of promoter elements, each with their own corresponding sigma factor, which allows for orthogonal expression. By our knowledge, this is the first set of orthogonal promoter libraries that can be combined in *E. coli*. Orthogonal expression is a key characteristic in bioengineering and synthetic biology, as it allows for no interference with the host's physiology and thus reduce undesirable interactions. Both of the objectives are innovative, novel and of high interest to the bioengineering and synthetic biology community, where the selection of promoter elements is, at this moment, still limited to existing (limited) libraries holding well-characterized promoter sequences (e.g. Anderson promoter collection).

As requested by the reviewer, we have provided influential factors that determine the promoter strength in the promoter spacer. Results are depicted in Supplementary Figure 8-11, and are obtained using DeepLIFT. However, we want to remark that we refrained from making any interpretations, as we believe that insights/conclusions drawn from these DNA sequence motifs offer no guarantee that they

apply to any promoter sequence outside of this scope. Importantly, the spacer has been mapped using a conserved promoter region. Thus, the effect of variation outside the spacer sequence cannot be predicted. As such, variation of the non-spacer sequence could very well influence the importance of certain input features on the genome. The manuscript is modified accordingly:

Manuscript (end of Section 2.4)

Models predicting the promoter TIF of σ^F - and especially σ^W -specific promoters have relatively low performances (Table 1), resulting in a less clear ordinal character of the data, with a higher overlap between the distributions of the predicted labels from different classes. Using DeepLIFT, a sensitivity analysis of the input as a function of the output was performed on the trained model of each σ -specific promoter library (Supplementary Figure 8-11). This is done on all sequences in the test set. Attribution scores are calculated based on the gradient (backpropagated from the target class) and signify the relevance of the input nucleotides on the output class prediction.

Supplementary Files

Supplementary Figure 8: Attribution scores obtained by DeepLIFT for the model trained on σ^{70} -specific promoters. Data are represented as mean values for each sequence in the test set (n= 56,885 samples) and separated by class (following the class distributions listed in Supplementary Table 1) and position (x-axis).

σ^B library

Supplementary Figure 9: Attribution scores obtained by DeepLIFT for the model trained on σ^B -specific promoters. Data are represented as mean values for each sequence in the test set (n= 29,850 samples) and separated by class (following the class distributions listed in Supplementary Table 1) and position (x-axis).

σ^F library

Supplementary Figure 10: Attribution scores obtained by DeepLIFT for the model trained on σ^F -specific promoters. Data are represented as mean values for each sequence in the test set ($n=44,864$ samples) and separated by class (following the class distributions listed in Supplementary Table 1) and position (x-axis).

σ^W library

Supplementary Figure 11: Attribution scores obtained by DeepLIFT for the model trained on σ^W -specific promoters. Data are represented as mean values for each sequence in the test set ($n= 36,963$ samples) and separated by class (following the class distributions listed in Supplementary Table 1) and position (x-axis).